# Single-cell analysis of human primary prostate cancer reveals the heterogeneity of tumor-associated epithelial cell states

Hanbing Song [1,2,3,4], Hannah N. W. Weinstein [1,2,3,4,15], Paul Allegakoen[1,2,3,4,15], Marc H. Wadsworth II[5,6,7,8,9,15], Jamie Xie[1,2,3,4], Heiko Yang[2,10], Ethan A. Castro[1,2,3,4], Kevin L. Lu[11], Bradley A. Stohr[11], Felix Y. Feng [2,10,12], Peter R. Carroll[2,10], Bruce Wang [13], Matthew R. Cooperberg[2,10,14], Alex K. Shalek [5,6,7,8,9] & Franklin W. Huang [1,2,3,4,14✉]

Prostate cancer is the second most common malignancy in men worldwide and consists of a mixture of tumor and non-tumor cell types. To characterize the prostate cancer tumor microenvironment, we perform single-cell RNA-sequencing on prostate biopsies, prostatectomy specimens, and patient-derived organoids from localized prostate cancer patients. We uncover heterogeneous cellular states in prostate epithelial cells marked by high androgen signaling states that are enriched in prostate cancer and identify a population of tumor-associated club cells that may be associated with prostate carcinogenesis. *ERG*-negative tumor cells, compared to *ERG*-positive cells, demonstrate shared heterogeneity with surrounding luminal epithelial cells and appear to give rise to common tumor micro-environment responses. Finally, we show that prostate epithelial organoids harbor tumor-associated epithelial cell states and are enriched with distinct cell types and states from their parent tissues. Our results provide diagnostically relevant insights and advance our under-standing of the cellular states associated with prostate carcinogenesis.

[1] Division of Hematology/Oncology, Department of Medicine, University of California, San Francisco, San Francisco, CA 94143, USA. [2] Helen Diller Family Comprehensive Cancer Center, University of California, San Francisco, San Francisco, CA 94143, USA. [3] Bakar Computational Health Sciences Institute, University of California, San Francisco, San Francisco, CA 94143, USA. [4] Institute for Human Genetics, University of California, San Francisco, San Francisco, CA 94143, USA. [5] The Ragon Institute of Massachusetts General Hospital, Massachusetts Institute of Technology and Harvard University, Cambridge, MA 02139, USA. [6] Institute for Medical Engineering and Science (IMES), Massachusetts Institute of Technology, Cambridge, MA 02139, USA. [7] Department of Chemistry, Massachusetts Institute of Technology, Cambridge, MA 02139, USA. [8] Koch Institute for Integrative Cancer Research, Massachusetts Institute of Technology, Cambridge, MA 02139, USA. [9] Broad Institute of Massachusetts Institute of Technology and Harvard, Cambridge, MA 02142, USA. [10] Department of Urology, University of California, San Francisco, San Francisco, CA 94143, USA. [11] Department of Pathology, University of California, San Francisco, San Francisco, CA 94143, USA. [12] Departments of Radiation Oncology, University of California, San Francisco, San Francisco, CA 94143, USA. [13] Division of Gastroenterology, Department of Medicine, University of California, San Francisco, CA 94143, USA. [14] Division of Hematology and Oncology, Department of Medicine, San Francisco Veterans Affairs Medical Center, San Francisco, CA 94121, USA. [15] These authors contributed equally: Hannah N. W. Weinstein, Paul Allegakoen, Marc H. Wadsworth II. ✉email: Franklin.Huang@ucsf.edu

The prostate consists of multiple cell types, including epithelial, stromal, and immune cells, each of which has a specialized gene expression profile. The development of cancer from prostate tissue involves complex interactions of tumor cells with surrounding epithelial and stromal cells and can occur multifocally, suggesting that prostate epithelial cells may undergo cellular state transitions towards carcinogenesis[1–6]. Previous studies on prostate cancer (PCa) molecular changes have focused on unsorted bulk tissue samples, leaving a gap in our understanding of the adjacent epithelial cell states.

The classification of prostate epithelial cells has been expanded over the past few years from three types (basal epithelial cells, luminal epithelial cells, and neuroendocrine cells)[7,8] to include hillock cells and club cells[9]. The roles of these additional cell types in the prostate are largely unknown. Most PCa are marked by the expansion of malignant cells with luminal epithelial features and the absence of basal epithelial cells. However, to date, the role of additional cell populations beyond the luminal and basal types is not well known.

Another underexplored area is the tumor microenvironmental changes that occur based on dominant genomic drivers in PCa. PCa tumor cells are driven by a number of oncogenic alterations including highly prevalent gene fusion events such as *TMPRSS2-ERG* and others involving ETS family transcription factors such as *ETV1/4/5*[1,10–12]. Tumor cells without ETS-fusion events and non-malignant luminal cells, however, have not been thoroughly characterized on a single-cell level, and uncertainty remains whether ETS-fusion events could evoke differential stromal and immune cell responses.

Here, we analyze at single-cell resolution the tumor microenvironment and cellular states associated with prostate carcinogenesis in localized prostate cancer samples. We characterize tumor cells and the surrounding epithelial, stromal, and immune cell microenvironment and identify cell states that are associated with tumorigenesis via single-cell RNA-sequencing (scRNA-seq). Furthermore, using in vitro organoids from PCa tumor tissues, we describe molecular and cellular features of prostate epithelial organoids compared to prostate tissues.

## Results

To probe the diversity of cell types and transcriptional states of cells in localized prostate cancer specimens, we obtained prostate cancer tissue from transrectal prostate biopsies and radical prostatectomy (RP) specimens from men with localized prostate cancer ($N = 11$ patients, Supplementary Data 1). Single cells were isolated for scRNA-seq (Supplementary Data 1) using an improved Seq-well single-cell platform[13]. Altogether, 21,743 cells were analyzed and a total of 9 different major cell types were identified, marked by specific gene expression profiles ("Methods", Fig. 1a, b).

Cell-type identification was determined by examining differentially expressed genes (DEGs) as well as signature scores from normal prostate and immune cell population gene sets[9,14]. Cells in the merged dataset were annotated as epithelial, stromal (endothelial, fibroblast, and smooth muscle) and immune cells (T-cells, myeloid cells, plasma cells, mast cells, and B cells) based on established marker genes. Epithelial cells ($N = 13,322$) were identified based upon the expression of luminal epithelial (LE) markers *KLK3, ACPP,* and *MSMB*, consistent with LE cells found as the dominant epithelial cell type in PCa samples. Immune cells were identified based on the high-level expression of *PTPRC* in five clusters, of which one cluster was marked by high-level expression of *IL7R, CD8A,* and *CD69*, indicating a mixture of both CD8 and CD4 T-cells; a second cluster was characterized by the myeloid cell markers *APOE, LYZ,* and *IL1B*[15–18]. The third

*PTPRC* + cluster represented plasma cells marked by high-level expression of *MZB1* and *IGJ*. The other two remaining *PTPRC* + clusters were annotated as mast cells expressing *CPA3, KIT,* and *TPSAB1*, and a population of B cells expressing *MS4A1, CD22,* and *CD79A*. Stromal cells consisted of endothelial cells characterized by *CLDN5* and *SELE* expression, fibroblasts expressing *C1S, DCN,* and *C7*, and smooth muscle cells expressing *ACTA2, MYH11,* and *RGS5* (Fig. 1c).

As our samples consisted of prostate biopsies ($N = 3$ patients) and RP specimens ($N = 8$ patients), half of which had matched benign-appearing tissue (Supplementary Data 1), we tested whether each sampling strategy captured a similar distribution of different cell types across samples. All major cell types were captured in each sample with epithelial cells comprising the largest population (Fig. 1d). No significant difference was found among the three sample types ($P > 0.05$, Mann–Whitney $U$ test) (Fig. 1e). We also compared the cell-type composition of epithelial cells, stromal cells (endothelial, fibroblasts, and smooth muscle), and immune cells (T cells and myeloid cells) among paired tumor ($N = 4$), paired normal ($N = 4$), and RP unpaired tumor tissues ($N = 4$) (Supplementary Data 1) and found no significant differences. The main cell types identified were validated by SingleR annotation[19] (Supplementary Data 1). Furthermore, within each biopsied patient, we tested whether biopsies from the two anatomical regions identified similar cell types and found that all cell types were recovered in each biopsy sample with some sampling differences by anatomical regions (Supplementary Data 1).

**Epithelial cell clusters reveal tumor cells and surrounding non-tumor epithelial cell heterogeneity**. To identify the transcriptional cell states of epithelial cells associated with prostate cancer, we performed a graph-based clustering analysis and identified 20 clusters (Fig. 2a). We then conducted single-sample gene-set enrichment analysis[20,21] (ssGSEA) using signature gene sets developed from single-cell profiling of normal prostates (Supplementary Data 2) in a previous study to determine the major cell subtypes[9]. Clusters with *KRT5, KRT15, KRT17,* and *TP63* expression (Fig. 2b) and significantly upregulated basal epithelial (BE) signature scores were identified as BE cells. Given that tumor cells predominantly express LE cell markers such as *KLK2, KLK3, ACPP,* and *NKX3-1*, clusters with high LE signatures scores could be either tumor cells or non-malignant LE cells (Fig. 2b). BE and LE signature feature plots also revealed a cluster of cells (cluster 5) that we termed other epithelial (OE) cells (Fig. 2a, c) with lower BE and LE signatures scores (Supplementary Fig. 1a) and were characterized by several markers previously identified as associated with PCa including *PIGR, MMP7* and *CP* (Fig. 2b). In previous studies, *PIGR* has shown a role in promoting cell transformation and proliferation[22], *MMP7* may promote prostate carcinogenesis through induction of epithelial-to-mesenchymal transition[23], and serum *CP* levels have been used as a marker in PCa[24].

Other than these three major types of epithelial cells, we next aimed to identify the putative tumor cells within our dataset. Approximately 50% of PCa tumors from European ancestry patients harbor *TMPRSS2-ERG* fusion events and less frequently harbor other ETS-fusion events (*ETV1, ETV4, ETV5*)[25]. Therefore, we tested cells for *ERG, ETV1, ETV4,* or *ETV5* expression, and found that *ERG* expression was upregulated in four clusters which we annotated as *ERG*-positive (*ERG*+ ) tumor cells (Fig. 2b and Supplementary Fig. 1a) but no cluster showed expression of *ETV1, ETV4,* or *ETV5*, suggesting that the six patients that contributed to these four *ERG*+ tumor cells harbored *ERG* fusion events. The identity of *ERG*+ tumor cells

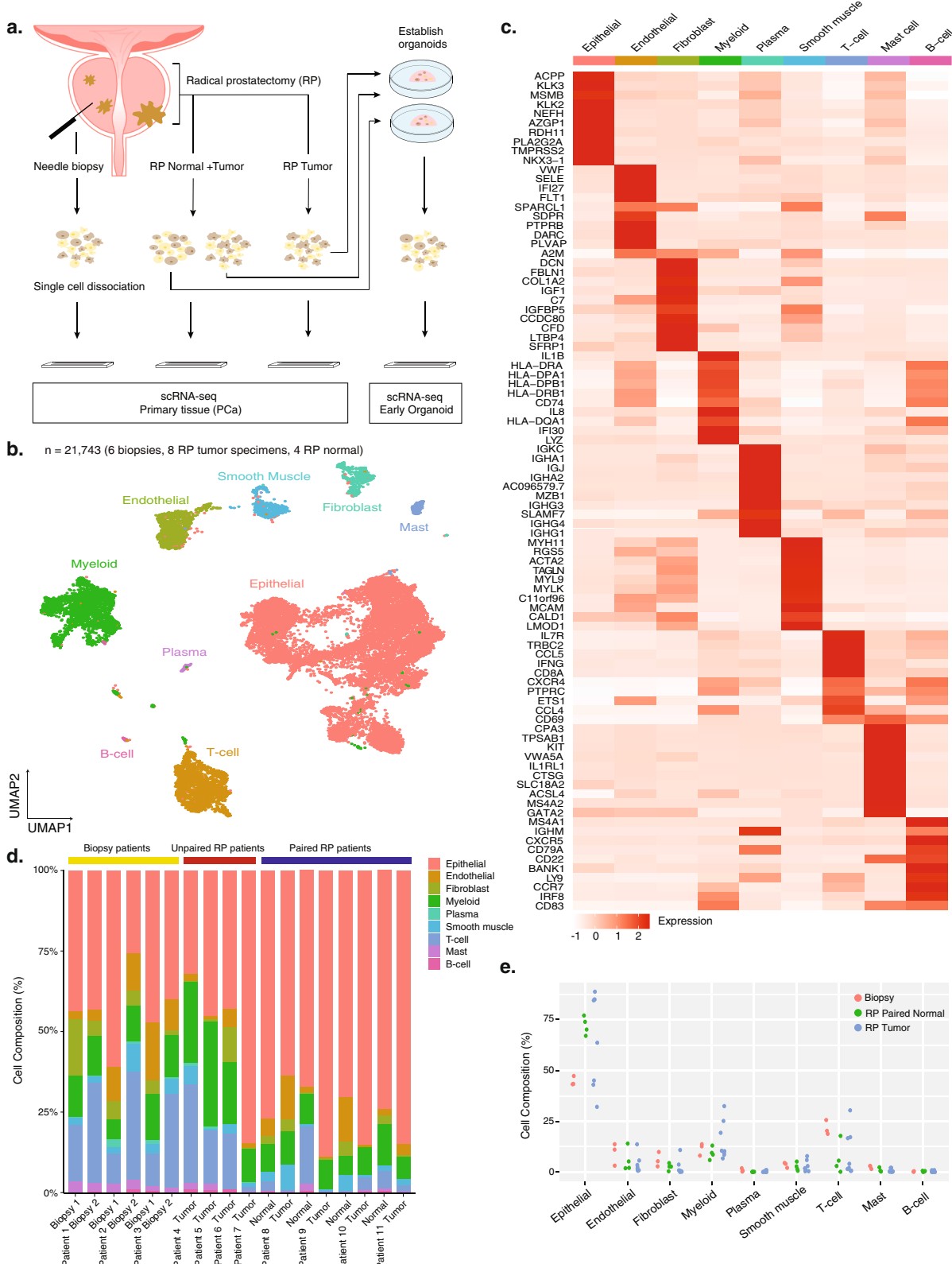

was further supported by the upregulation of the SETLUR PROSTATE CANCER TMPRSS2-ERG FUSION UP gene-set signature score in these cells[26] (Supplementary Fig. 1b). *ERG* status was histologically validated in a subset of patients using immunohistochemistry (Supplementary Fig. 2b). Furthermore, STAR-Fusion[27] identified potential fusion transcripts of *TMPRSS2-ERG* fusions in two *ERG+* patients.

To identify tumor cells without ETS-fusion events, we tested the LIU PROSTATE CANCER UP and other known PCa tumor marker gene-set signature scores and identified 11 clusters in total with upregulated signature scores of at least one prostate cancer gene set (Supplementary Fig. 1a, b). Single-sample gene-set enrichment analysis (ssGSEA) on these 11 clusters also showed at least one prostate cancer gene set that scored in the top 1% of all

**Fig. 1 Prostate cancer (PCa) sample single-cell RNA-sequencing overview and identification of major cell types in localized prostate cancer. a** Single-cell RNA-sequencing workflow on PCa biopsies, radical prostatectomy (RP) specimens, and in vitro organoid cultures grown from RP tumor specimens using the Seq-Well platform. **b** Overview of major cell types identified within the combined dataset consisting of 21,743 cells from all biopsies ($N = 6$) and RP specimens ($N = 12$). Cell types are labeled in colors from corresponding clusters in the Uniform Manifold Approximation and Projection (UMAP). **c** Heatmap for the top ten differentially expressed genes in each cell type. **d** Cell-type composition stacked bar chart by sample. Cell counts for each sample are normalized to 100%. Sample type is annotated (top) and patients are labeled below the x axis. **e** Cell composition comparison for each cell type among three sample types: biopsy patients ($N = 3$), RP tumor specimens ($N = 8$), and RP paired normal tissues ($N = 4$). Each sample type is represented by a different color. Source data are provided as a Source Data file.

C2CGP gene-set collection ($N = 3297$) (Supplementary Data 3). These 11 clusters included the four *ERG*+ tumor cell clusters we previously annotated and seven other clusters with no *ERG* expression which we annotated as *ERG*-negative (*ERG*-) tumor cells (Fig. 2c). The *ERG*− tumor cell clusters were characterized by the enrichment of at least one known PCa gene-set signature and higher expression of tumor markers such as *SPON2*[28] and *PCA3* compared to non-tumor LE cells in our dataset (Fig. 2b). No tumor clusters detected within our dataset showed enrichment of a BE signature.

To validate our tumor cell assignments, we estimated copy number variations (CNV) with InferCNV[29], using non-malignant LE cells as a reference. From the CNV estimation visualization (Supplementary Fig. 1c), we identified significantly different CNV patterns in both *ERG*+ and *ERG*− tumor cells. Non-uniform CNV profiles were detected within *ERG*+ and *ERG*− tumor cell populations, suggesting heterogeneity in both tumor cell populations. We also performed whole-exome sequencing (WES) for three RP samples to confirm tumor content. In the WES samples, we detected somatic mutations in known PCa genes, including *KMT2D, MTOR, SPOP*, and *PIK3R1* (Supplementary Data 4), indicating tumor content in tissues assessed by single-cell analysis. Copy number analysis of the WES samples revealed similar CNV events with the inferCNV estimation, such as chr4q31 amplification, chr11q24 amplification, and chr19q13 deletion (Supplementary Data 4), supporting our tumor cell identification.

While we did not observe a separate neuroendocrine cell cluster, we tested for prostate neuroendocrine (NE) cells[9,30] using an established NE cell signature gene set[9] and computed the NE signature scores for each epithelial cell. Taking the cells ranking in the top 0.5% NE signature score, we detected 66 putative NE cells within the BE cell population, characterized by *CHGB, KRT4*, and *LY6D* expression[9] (Supplementary Fig. 1c, d).

To examine if our annotation method could accurately identify each epithelial cell type, we computed the top ten biomarkers for each cell type (Fig. 2d). BE cells showed high expression of established basal epithelial cell markers *KRT5, KRT15*, and *KRT17*. The top biomarkers in the OE clusters were *PSCA, PIGR, MMP7, SCGB1A1*, and *LTF*, of which *PSCA* is upregulated in PCa[31–33] and *SCGB1A1* is a marker for lung club cells[34]. *ERG*+ and *ERG*− tumor cells and non-malignant LE cells all showed high expression of luminal markers *KLK3, KLK2*, and *ACPP*[35]. *ERG*+ tumor cells were characterized by expression of *ERG* and tumor markers *PCA3, AMACR*, and *TRPM8*;[35–37] *ERG*− tumor cells were marked by the expression of tumor markers *PCA3* and *TRPM8*[35–37] (Fig. 2d).

Since most PCa is androgen-responsive with tumor cell proliferation dependent on the activity of the androgen receptor (*AR*)[36–39], we tested for androgen responsiveness among the epithelial cell populations and identified LE cells and tumor cells as the most androgen-responsive as they scored significantly higher than other epithelial cell types in *AR* signature scores (Supplementary Fig. 1a). To identify putative prostate cancer stem cells that may contribute to prostate cancer development, we

used an adult stem cell signature gene set[38] and found that 56.4% of the BE cell population was enriched for the stem cell signature (Supplementary Fig. 1b).

A previous single-cell study of normal human prostate reported two populations of other epithelial cells: hillock cells characterized by *KRT13, SERPINB1, CLDN4*, and *APOBEC2* expression and club cells characterized by the expression of *SCGB3A1, PIGR, MMP7, CP*, and *LCN2*[9]. While we did not detect a separate hillock cell population within our prostate cancer epithelial cells (Supplementary Fig. 1e), we did detect a distinct population representing 6.5% of all epithelial cells (872 of 13,322) characterized by expression of *PIGR, MMP7, CP*, and *LTF* (Fig. 2d) (FDR $q < 10e-20$). We hypothesized that this epithelial cluster represented club cells that had previously been described in lung[34] and normal prostate specimens[9]. To test this hypothesis, we applied a normal prostate club cell gene set signature[9] and projected it onto our epithelial Uniform Manifold Approximation and Projection (UMAP). We found that cells with high club cell signature scores largely overlapped with this OE cluster (cluster 5) (Fig. 2e). Furthermore, this cluster was enriched for a lung club signature compared to other clusters ($p < 2.2e-16$, Wilcoxon rank-sum test) (Fig. 2f). Based on these results, we annotated this cluster as club cells. We then conducted a ssGSEA analysis on all epithelial cells using the BE, LE, and club cell signatures generated from the DEG profiles (Supplementary Data 2). All three cell-type signature scores were strongly correlated to the corresponding cell types, supporting our annotation (Supplementary Fig. 1f).

**Club and BE cells harbor PCa-enriched LE-like cell states that are upregulated in AR signaling.** A recent study identified a luminal progenitor cell type in mouse and human prostates characterized by high expression of LE markers *KRT8, KRT18*, and other markers including *PSCA, KRT4, TACSTD2*, and *PIGR*[39]. In epithelial cells from both paired normal and tumor samples in our study, we could not identify a distinctive cell cluster by the co-expression of *KRT8, KRT18*, and *TACSTD2*; however, the club cell population we identified was characterized by higher *PSCA* and *PIGR* expression compared to other epithelial cell types (Supplementary Fig. 3a).

Club cells in PCa have not been previously characterized. Since we exclusively captured club cells but not hillock cells in our PCa samples, we hypothesized that club cells may be associated with carcinogenesis and PCa-club cells might be enriched in certain cell states. To test this hypothesis, we integrated our prostate cancer club cells (Club PCa) with normal club cells from a previous study from healthy controls[9] (Club Normal) and detected six cell states with distinct transcriptomic profiles (Fig. 3a) by selecting an optimal resolution to yield stable clusters (Supplementary Fig. 3b). Overall, compared to club cells from normal samples, PCa-club cells exhibited downregulation of genes including lipocalin 2 (*LCN2*) and a growth-inhibitory cytokine *SCGB3A1*[40,41] and upregulation of *LTF, AR*, and *AR* downstream members including *KLK3, KLK2, ACPP*, and *NKX3-1* (Fig. 3b), which we hypothesized could be driven by the

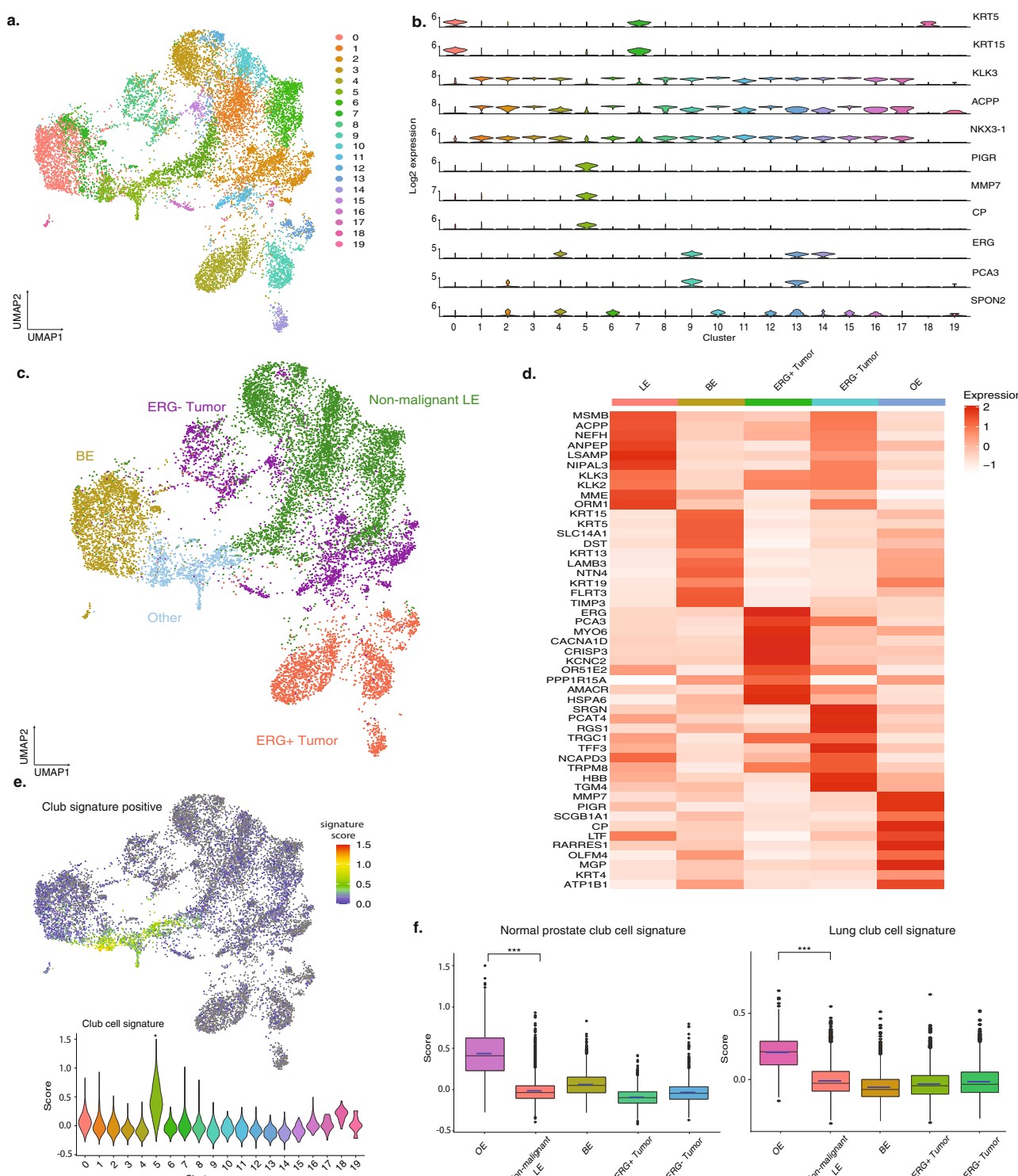

**Fig. 2 Identification of tumor cells and major epithelial cell types, including club cells. a** UMAP projection of all 20 clusters identified in the epithelial cells. Clusters are labeled in the UMAP. **b** Violin plots of representative marker genes across the clusters. **c** UMAP of epithelial cells annotated by cell types. **d** Heatmap of the top ten differentially expressed genes in each cell type (BE: basal epithelial cells; Other: other epithelial cells; Non-malignant LE: non-malignant luminal epithelial cells; *ERG + Tumor*: *ERG +* tumor cells; *ERG−* Tumor: *ERG−* tumor cells). **e** Club cell signature scores of each epithelial cell projected on the UMAP and signature score violin plots across all clusters. **f** Box plots of club cell signature scores from normal club cells and lung club cells across epithelial cell types (*N* = 13,322 cells, ***P* < 0.001, Wilcoxon rank-sum test; normal club cell signature: *P* < 2.2e-16; normal club cell signature: *P* < 2.2e-16). Center, bounds, and percentiles are shown in the box plot. Source data are provided as a Source Data file.

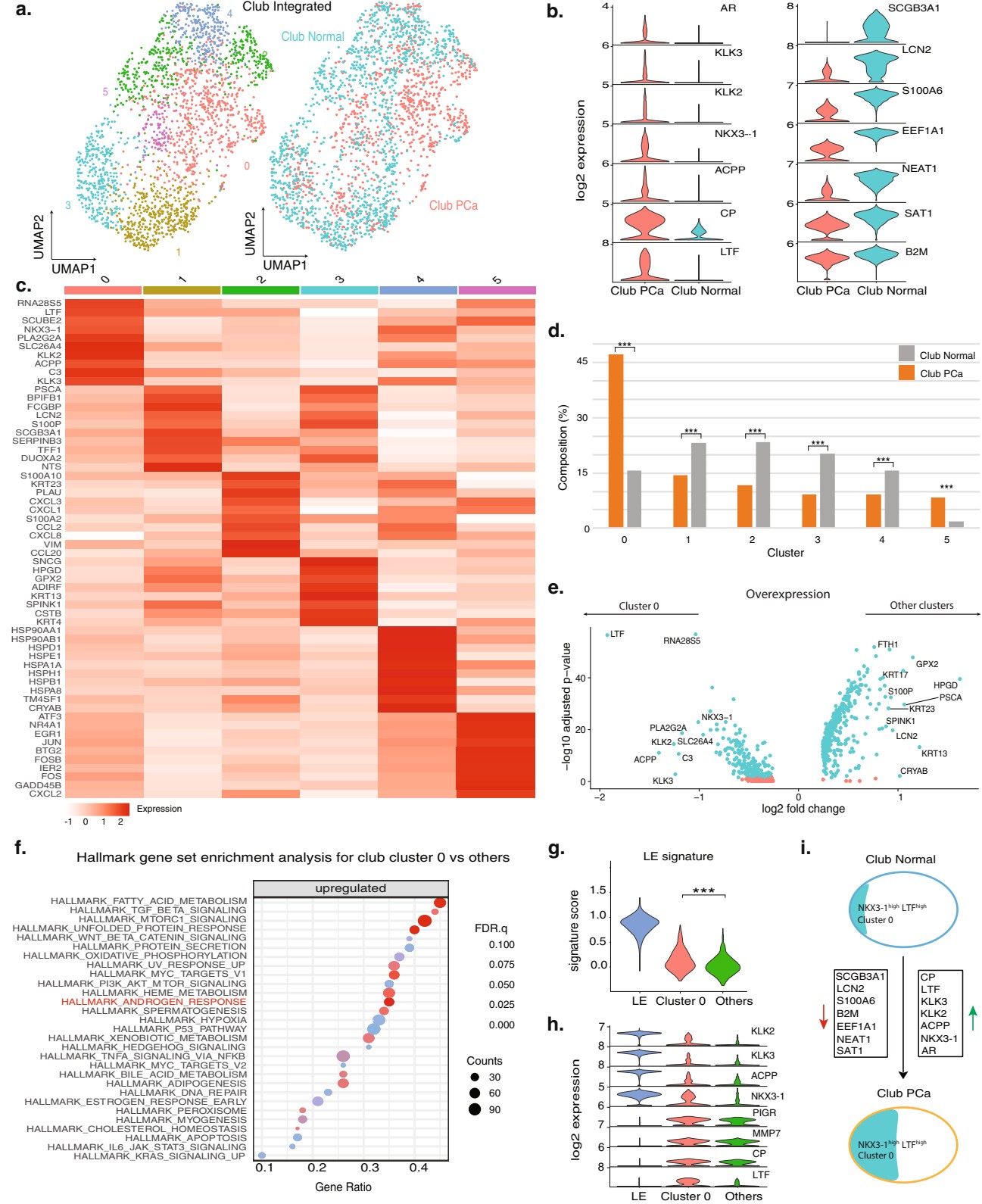

enrichment of one or more specific club cell states in the PCa samples.

For each of the six subclusters, a group of distinctive DEGs was identified (Fig. 3c) and each subcluster was detected in both Club PCa and Club Normal (Supplementary Fig. 3c), of which, cluster 0 represented over 45% of Club PCa and was enriched in Club PCa by more than three folds (*P* < 0.001, two-sided Fisher's exact

test (FET)) (Fig. 3d). This cluster was distinguished by a higher level of expression of *LTF*, luminal markers, and downstream *AR* pathway genes *KLK2*, *KLK3*, *ACPP*, *PLA2G2A*, and *NKX3-1* (Fig. 3e), suggesting a luminal-like and androgen-responsive state[39].

To test the functional role of this cell state in silico, we performed GSEA analysis using C2 canonical pathways

**Fig. 3 Identification of PCa-enriched club cell states with upregulated androgen response signature. a** UMAP of integrated club cells from PCa samples (Club PCa) and club cells from normal samples (Club Normal), color-coded by cell states with differential gene expression profiles (left) and sample type (right). **b** Violin plots of representative marker genes between the two types of club cells. **c** Heatmap for the top ten differentially expressed genes in each cell state. **d** Grouped bar chart comparison of six cell-state compositions between Club PCa and Club Normal. Significance levels are labeled (***$P < 0.001$, two-sided Fisher's exact test; cluster 0: $P = 7.21e{-}58$; cluster 1: $P = 3.11e{-}07$; cluster 2: $P = 2.72e{-}12$; cluster 3: $P = 1.05e{-}12$; cluster 4: $P = 7.36e{-}06$; cluster 5: $P = 1.20e{-}12$). **e** Volcano plots of the overexpressed genes in Club cell cluster 0 and other cell states within the PCa samples. **f** Top 20 upregulated signaling pathways between Club cell cluster 0 and the other club cells on Hallmark gene-set collection ($N = 50$) within the PCa samples. Gene counts for the corresponding gene set are indicated by marker radius. Statistical significance levels (FDR) are shown by the color gradient. **g** Comparison of LE signature scores between Club cluster 0 and other club cells (***$P = 2.33e{-}19$, two-sided Wilcoxon rank-sum test), and between within the PCa samples. **h** Violin plot comparison between Club cluster 0, other club cells, and LE for multiple LE and club cell markers within the PCa samples. **i** Schematic marker of gene expression changes between Club Normal and Club PCa. Gene downregulation and upregulation in Club PCa compared to Club Normal are represented by red and green arrows. The proportion of Club cell cluster 0 within all club cells represented by the area in blue and characterized by its LE-like state and high-level expression of *LTF* and *NKX3-1*. Source data are provided as a Source Data file.

($N = 2232$) (Supplementary Data 5) and Hallmark ($N = 50$) gene set *collections* on cluster 0 vs other cell states. Among the top significantly upregulated gene sets in cluster 0 was the Hallmark Androgen Response pathway (FDR $q < 10e{-}5$) (Fig. 3f). These results were consistent with the upregulation of downstream *AR* pathway genes in cluster 0.

Next, we tested whether this PCa-enriched cell state represented a luminal-like cell state. We observed higher LE signature scores in cluster 0 compared to other cell states ($P < 0.001$, Wilcoxon rank-sum test) (Fig. 3g). Specifically, we compared the expression levels of both LE and club cell markers among cluster 0, other club cells, and the LE population within the PCa samples, and found that club cell cluster 0 exhibited higher expression of LE markers *KLK2, KLK3, ACPP*, and *NKX3-1* than other cell states (Fig. 3h) while *AR* itself was not upregulated in cluster 0 (Supplementary Fig. 3d), and that expression of club markers *PIGR, MMP7, CP*, and *LTF* in all club cells was significantly higher than in the LE population (Fig. 3h). Overall, the population of PCa-club cells, compared to normal prostate clubs, was characterized by higher androgen signaling and enrichment of an *LTF*[high] and *NKX3-1*[high] luminal-like cell state (Fig. 3i).

The finding of a luminal-like club cell state led us to investigate if a similar cell state existed in the BE cell population of prostate cancer samples. Therefore, we integrated BE cells in the PCa samples (BE PCa) with BE cells from normal samples (BE Normal) and identified 9 cell states (Fig. 4a and Supplementary Fig. 4a) with distinctive DEGs (Supplementary Fig. 4b). While all 9 cell states were represented in both BE PCa and BE Normal cells (Fig. 4b), cluster 6 was found to be significantly enriched in PCa samples (31.8% vs 0.2%, PCa vs Normal, $P < 2.2e{-}16$, two-sided FET) while cluster 4 was enriched in normal samples (0.8% vs 15.9%, PCa vs Normal) (Fig. 4b and Supplementary Fig. 4c). Cluster 6 was predominantly found in PCa samples (94.2% in BE PCa) and was marked by higher expression downstream *AR* pathway members *KLK3, KLK2*, and *ACPP* (Supplementary Fig. 4b). Compared to other BE cells in PCa samples, BE cluster 6 also showed significant upregulation of *AR* ($P = 0.004$, Wilcoxon rank-sum test, Supplementary Fig. 4d). Among the top significantly upregulated gene sets were the Hallmark Androgen Response pathway within the Hallmark gene set collection (Supplementary Data 5) as well as androgen response pathways, estrogen pathways, insulin signaling pathway, and Kegg pathways in cancer within the C2CP gene set collection (FDR $q < 0.1$, Wilcoxon rank-sum test) (Fig. 4d)[42–44]. As *AR* pathway members were among the top biomarkers for cluster 6 (Fig. 4e), we hypothesized that BE cluster 6 may represent an intermediate BE/LE cell state, even though it did not cluster separately from other BE cells on the epithelial cell UMAP (Fig. 4f). Therefore, we compared the expression levels of BE and LE markers between BE cluster 6 and other BE cells, and found that the expression of

basal markers such as *KRT5, KRT15*, and *TP63* did not show any significant differences. The only significant difference in BE cluster 6 was the higher expression of luminal markers (Fig. 4g), though at lower levels compared to the PCa LE cell population. Moreover, we found that BE cluster 6 was significantly upregulated in the Hallmark Androgen Response signature and LE signature score ($P < 0.001$, Wilcoxon rank-sum test, Fig. 4h), supporting that this cell state may be a luminal-like state associated with prostate cancer.

Similarly, we identified eight cell states within the integrated LE dataset (Supplementary Fig. 4e). Unlike BE and club cells, we observed a clear separation between LE PCa and LE Normal (Supplementary Fig. 4e). Four cell states were found to be significantly enriched in LE PCa and two in LE Normal ($P < 0.001$, two-sided FET) (Supplementary Fig. 4f). Cluster 5 was marked by co-expression of club cell markers such as *PIGR, MMP7*, and *CP*, suggesting an intermediate population of LE and club cells. Cluster 1 was characterized by the overexpression of the *AR*-regulated gene *TMEFF2* and insulin-like growth factor *IGFBP5* compared to other cell states, and cluster 2 was upregulated in *AR* expression (Supplementary Fig. 4g).

We then tested if the PCa-enriched cell states in BE and club cells (Club cell cluster 0 and BE cluster 6) could be distinguished from other cell states in the differentiation trajectory. Using BE cells as the starting point consistent with a previous prostate scRNA-seq study[9], we plotted the diffusion pseudotime trajectory and observed that tumor cells and LE cells (*KLK3+*) were later than club cells (*PIGR+, LTF+*, and *PSCA+*) and *KRT5+* BE cells (Supplementary Fig. 5a, b), and that there was no significant difference in the computed pseudotime between BE and club cells (Supplementary Fig. 5c). We repeated the pseudotime analysis in an unsupervised manner without specifying the starting point and the pseudotime trajectory was consistent that BE cells give rise to club cells, LE, and both *ERG+* and *ERG−* tumor cells in our dataset. Then, we projected the epithelial cells on the partition-based graph abstraction (PAGA) embedding with a list of cell-type-specific markers (Supplementary Fig. 5d), showing distinctive transcriptomic profiles in each cell population.

To further investigate our finding of PCa-related club cells and the club cell state with upregulated AR signaling, we re-analyzed two publicly available datasets of PCa scRNA-seq samples[45,46]. In the Karthaus dataset ($N = 8$ patients), we observed that the luminal-2 population ($N = 10,603$ of 44,756 cells) was significantly enriched for our club cell signature (Supplementary Fig. 5e). Sub-clustering analysis of the luminal-2 population supported the presence of PCa-enriched club cell states with an upregulated luminal cell signature (Supplementary Fig. 5f). Similarly, in the Chen dataset ($N = 13$ patients), we tested our club cell signature gene set in the population originally annotated as BE_Intermediate population ($N = 1075$ of 36,424 cells) and

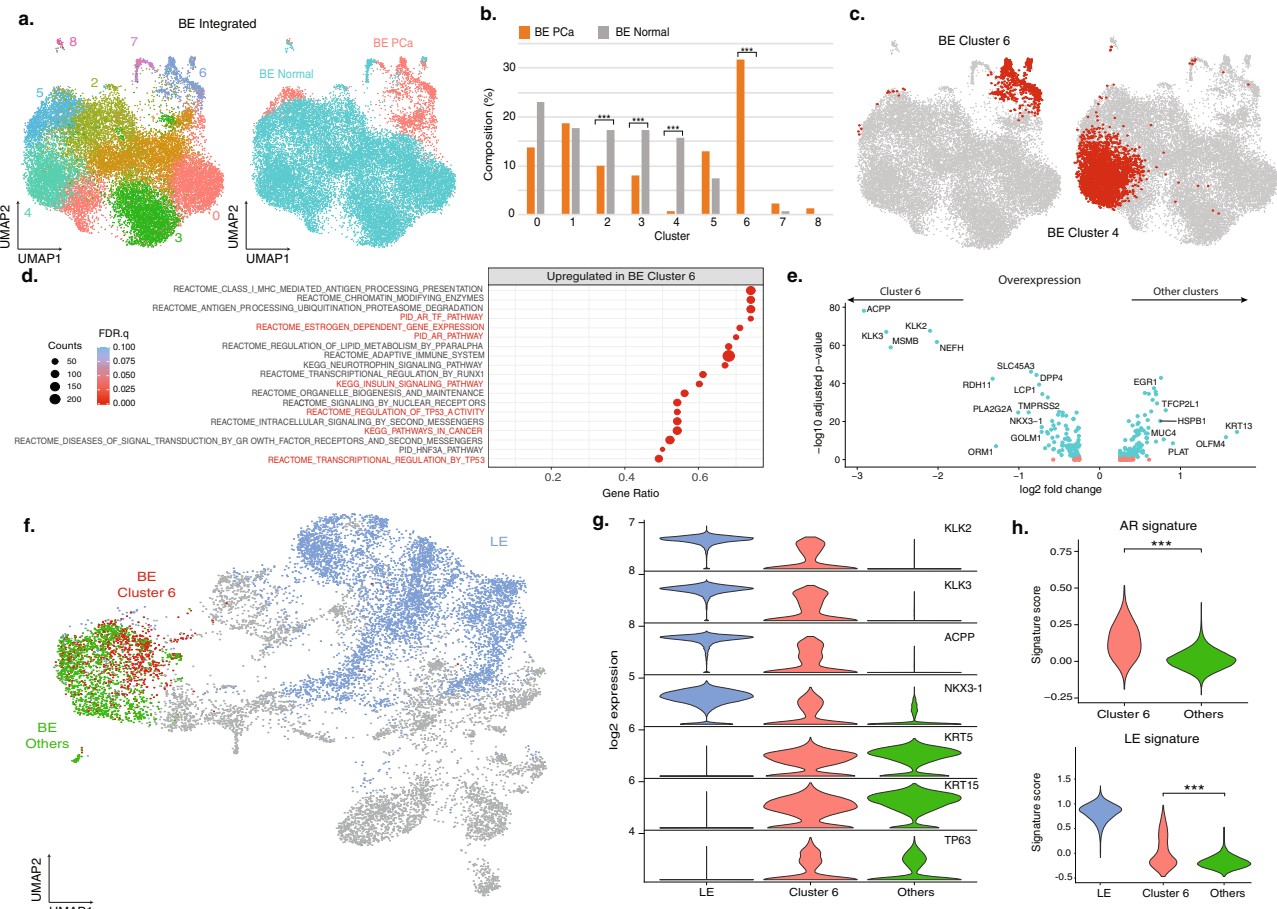

**Fig. 4 Integration of BE and LE cells identifies tumor-associated cell states enriched in the PCa samples. a** UMAP of integrated BE cells labeled by cell states (left) or samples type (BE PCa and BE Normal) (right). **b** Cell composition comparison between BE PCa and BE Normal (***$P < 0.001$, two-sided Fisher's exact test; cluster 2: $P = 1.24e-19$; cluster 3: $P = 2.00e-31$; cluster 4: $P = 9.31e-122$; cluster 6: $P < 2.2e-16$). **c** PCa and normal enriched cell states 4 and 6 highlighted in the integrated BE UMAP. **d** Top 20 upregulated signaling pathways between cluster 6 and the other BE on C2 canonical gene set (C2CP) collection ($N = 2,332$). Gene counts for the corresponding gene set are indicated by marker radius. Statistical significance levels (FDR) are shown by the color gradient. Pathways associated with PCa tumor progression and invasiveness are highlighted in red. **e** Volcano plots of the overexpressed genes in BE cluster 6 and other BE cell states within the PCa samples. **f** Distribution of BE cluster 6, other BE and LE on the overall epithelial cell UMAP. **g** Violin plot comparison between BE cluster 6, other BE and LE for multiple LE and BE markers within the PCa samples. **h** Comparison of Hallmark AR pathway signature and LE signature scores within the PCa samples (***$P < 0.001$, Wilcoxon rank-sum test; Hallmark AR pathway signature: $P = 2.10e-111$; LE signature score: $P = 2.95e-39$). Source data are provided as a Source Data file.

identified cells consistent with PCa-club cells from our dataset (Supplementary Fig. 5g). Sub-clustering revealed that both PCa-enriched BE and club cell states were present (Supplementary Fig. 5h) in the Chen et al. dataset. Compared to the other epithelial cells in these two datasets, the luminal-2 population in the Karthaus et al. dataset and BE_Intermediate population in the Chen et al. dataset showed significantly higher expression of PCa-club cell markers *PIGR, MMP7, LTF,* and *CP* (Supplementary Fig. 5i). After sub-clustering analysis within these two populations, we observed expression of PCa-club cell-enriched markers *PIGR, LTF,* and *NKX3-1* in multiple subclusters (Supplementary Fig. 5f).

**Integrated epithelial cell analysis reveals upregulated AR signaling in PCa samples.** As PCa samples in this study included four paired tumor and normal samples, we tested if PCa-enriched cell states in BE, LE, and club cells were enriched in the surrounding epithelial cells of the PCa biopsies and in radical proctectomy tissue samples containing tumor cells. We compared the percentage composition of each BE and club cell state within

all BE and club cells in all five sample types, respectively (Normal, biopsy, RP paired tumor, RP paired normal, and RP unpaired tumor). The PCa-enriched cell states of Club cell cluster 0 and BE cluster 6 were similarly represented in the four paired tumor and normal samples ($P = 0.43$, Mann–Whitney $U$ test).

To characterize the overall epithelial cell transcriptional programs in PCa samples, we integrated all PCa epithelial cells (Epithelial PCa) with prostate epithelial cells from normal healthy controls from a previous analysis (Epithelial Normal)[9] (Fig. 5a). We identified differentially expressed genes between tumor and normal samples across all three major types of epithelial cells (LE, BE, and club cells). We found ATF transcription factors *FOS* and *JUN*, members of the EGFR pathway that mediate gene regulation in response to cytokines and growth factors[47], and prostate acid phosphatase (*PSAP*)[48] as commonly upregulated genes across these cell types (Fig. 5b). However, when comparing between paired tumor and normal samples, no significant expression differences were detected for these DEGs (Supplementary Data 6), suggesting that compared to normal prostate samples, epithelial cells in the paired normal tissues were more similar to those from paired tumor tissues taken from different anatomical regions

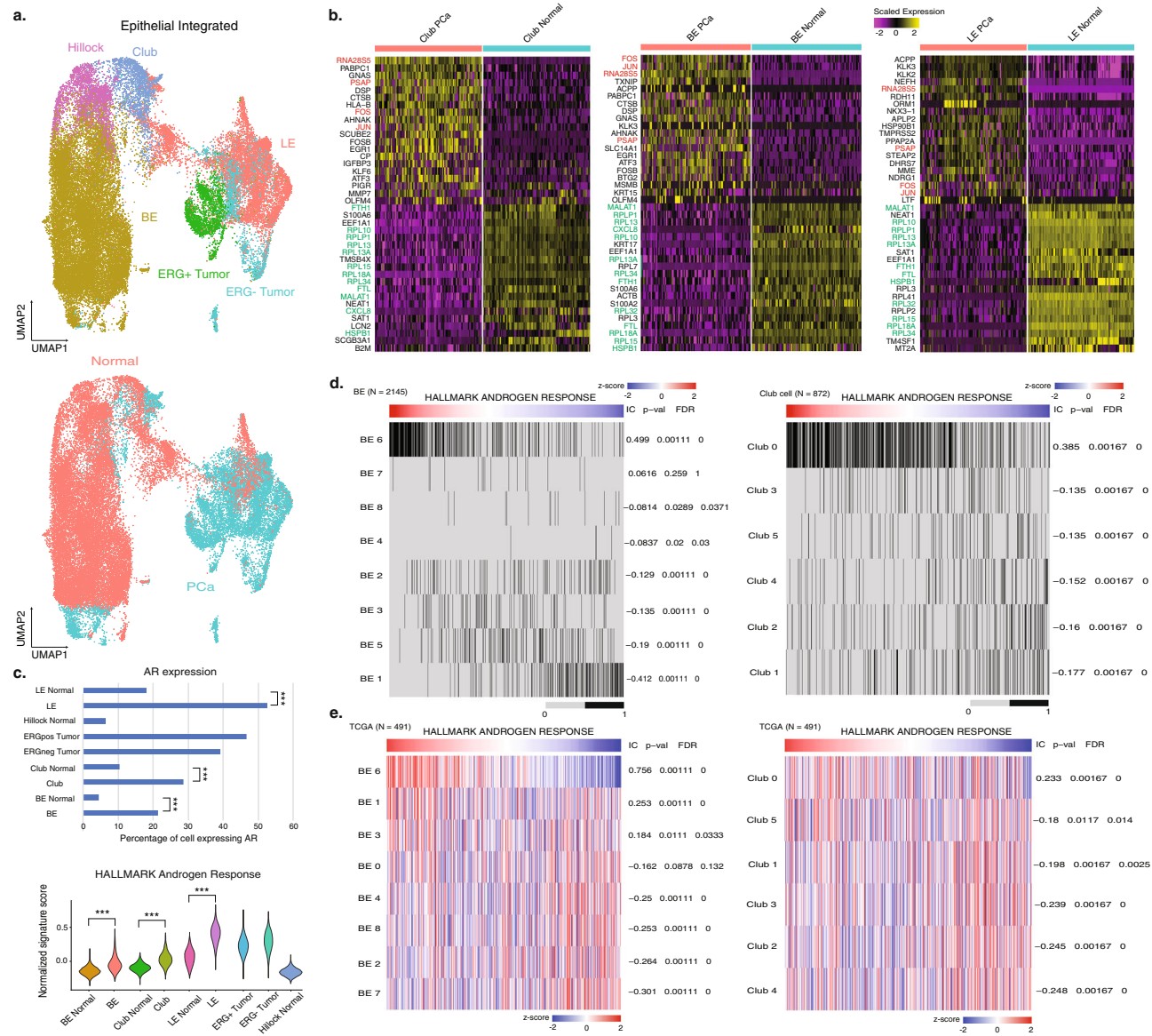

**Fig. 5 Integration of PCa and normal epithelial cells reveals common *AR* signaling upregulation driven by PCa-enriched BE and club cell states.**
**a** UMAP of integrated epithelial cells annotated by cell types and sample type (PCa and Normal), then separated by the origin (either previous normal epithelial cells or epithelial cells in the PCa samples). **b** Heatmaps of top 20 differentially expressed genes between PCa samples and normal prostates for adjacent cell types (left: BE PCa, BE Normal. Middle: Club Normal, Club PCa. Right: LE PCa, LE Normal). Commonly upregulated genes in the PCa samples are labeled in red, and commonly upregulated genes in the normal samples are labeled in green. **c** Top, *AR* expression percentages in all epithelial cell types within the integrated dataset. Significance levels are labeled in each comparison (***$P < 0.001$, two-sided Fisher's exact test; BE: $P = 1.49e-145$; LE: $P = 1.25e-184$; Club: $P = 1.61e-27$). Bottom, Comparison of Hallmark *AR* pathway signature scores of each epithelial cell type. Significance levels are labeled for each common cell type (***$P < 0.001$, Wilcoxon rank-sum test; BE: $P < 2.2e-16$; LE: $P = 4.21e-295$; Club: $P = 1.45e-253$). **d** The association of *AR* signature with BE and club cell state. Each cell is labeled (gray: 0, not in the cell state; black: 1, in the cell state). Information coefficient accompanied $P$ values and FDR $q$ values are labeled next to each cell state. **e** The association of *AR* signature with BE and club cell-state signature scores in the TCGA datasets ($N = 491$). Information coefficient accompanied $P$ values and FDR $q$ values are labeled next to each cell state. Source data are provided as a Source Data file.

within the same radical prostatectomy specimen. Since the two PCa-enriched cell states in BE and club cells showed upregulated *AR* signaling compared to other BE or club cells, respectively, we tested *AR* expression in the integrated dataset and found that in PCa epithelial cells, 21.4% of BE (458 of 2145 cells), 28.6% of club cells (249 of 872), 52.7% of LE (2974 of 5647 cells) and 43.2% of tumor cells (1993 of 4658 cells) were *AR*+, significantly higher compared to the same cell types from normal samples ($P < 0.001$, two-sided FET) (Fig. 5c). We also computed the Hallmark Androgen Response pathway signature scores for all cells and

found that the three major epithelial cell types in PCa samples were all upregulated in *AR* signaling compared to normal samples ($P < 0.001$, Wilcoxon rank-sum test) (Fig. 5c).

To validate the two PCa-enriched epithelial cell states we identified in BE and club cells and test their correlation with upregulated *AR* signaling, we ran ssGSEA analysis on all BE and club cells on the Hallmark Androgen Response pathway. The *AR* signature score of BE was only significantly positively correlated to BE cluster 6 (information coefficient (IC) = 0.499, FDR $q < 1e$-5), and the *AR* signature score in club cells was significantly

positively correlated to Club cell cluster 0 (IC = 0.385, FDR $q$ < 1e-5) (Fig. 5d). Furthermore, to test if this correlation between a PCa-enriched cell state and $AR$ signaling could be replicated in other PCa datasets, we projected all BE and club cell states across the TCGA[25] ($N$ = 499) and SU2C[49] ($N$ = 266) castration-resistant prostate cancer (CRPC) bulk RNA-seq datasets ("Methods"). In both bulk RNA-seq datasets, $AR$ signature scores were positively correlated with BE cluster 6 (IC = 0.756, FDR $q$ < 1e-5) and Club cell cluster 0 (IC = 0.233, FDR $q$ < 1e-5) (Fig. 5e), supporting our identification of cell states within BE cells and club cells that were more androgen-responsive and associated with prostate cancer.

**Transcriptomic profiles of $ERG+$ tumor cells are patient-specific while $ERG-$ tumor cells overlap with surrounding LE cells.** While $ERG+$ tumor cells clustered separately from non-malignant LE cells, $ERG-$ tumor cells resided more closely to non-malignant LE cells in the UMAP of all epithelial cells (Fig. 2c). To investigate this further, we first analyzed the sub-structure of $ERG+$ and $ERG-$ tumor cells separately to identify distinct underlying cell states (Fig. 6a, b). $ERG+$ tumor cells clustered in a patient-specific manner, whereas no such pattern was seen for $ERG-$ tumor cells as most $ERG-$ tumor cell states comprised more than one patient (Fig. 6c).

One possibility for the different distribution patterns between $ERG+$ and $ERG-$ tumor cells is that $ERG+$ tumor cells for each patient represented a distinctive cell state driven by a shared dominant oncogenic alteration, the $TMPRSS2-ERG$ fusion event, though no such distinction was seen in $ERG-$ tumor cells, suggesting more overlapping cell states between $ERG-$ tumor cells and adjacent non-malignant LE cells. To test this hypothesis, we integrated $ERG+$ tumor cells and $ERG-$ tumor cells separately with LE cells and performed sub-clustering analysis. Overall, we found 1244 genes significantly varied between $ERG+$ tumor cells and LE cells (FDR $q$ < 0.01, Wilcoxon rank-sum test) while only 314 genes were significantly varied between $ERG-$ tumor cells and LE cells (FDR $q$ < 0.01, Wilcoxon rank-sum test). Fourteen and seventeen cell states were recovered in the $ERG+$ and $ERG-$ integrated datasets, respectively (Supplementary Fig. 6a, b). We observed a clear separation between $ERG+$ tumor cells and non-malignant LE cells while $ERG-$ tumor cells were not clearly distinguishable from non-malignant LE in the analysis (Fig. 6d). From the cell-state composition comparison, we observed three cell states with more than 400 cells each that were almost exclusively detected in the $ERG+$ tumor cells, with each cell state largely attributed to one specific patient (Supplementary Fig. 6a). In contrast, no such patient specificity was observed for $ERG-$ tumor cells (Fig. 6e) (Supplementary Fig. 6b). In our dataset, $ERG+$ tumor cells were predominantly found in tumor samples while $ERG-$ tumor cells were found in paired tumor and normal samples (Supplementary Fig. 6c). Using the DEGs between $ERG+$ and $ERG-$ tumor cells (Supplementary Fig. 6d), we generated signature gene sets for both types of tumor cells and tested if the signatures of $ERG+$ and $ERG-$ tumor cells generated from this dataset were correlated with $TMPRSS2-ERG$ fusion status in TCGA[25] and SU2C[49] castration-resistant prostate cancer (CRPC) bulk RNA-seq datasets. $TMPRSS2-ERG$ fusion status was significantly positively correlated with an $ERG+$ tumor cell signature score in both datasets (TCGA: information coefficient (IC) = 0.673, FDR $q$ < 1e-5; SU2C: IC = 0.407, FDR $q$ < 1e-5) and the absence of $TMPRSS2-ERG$ fusion was significantly correlated with $ERG-$ tumor signature scores (TCGA: IC = −0.554, FDR $q$ < 1e-5; SU2C: IC = −0.211, FDR $q$ < 1e-5) (Fig. 6f). These results supported the tumor cell signatures and our use of $ERG$ expression as a classification in annotating tumor cells.

Furthermore, we compared the numbers of $ERG+$ tumor cell and $ERG-$ tumor cells in each patient. Tumor cells in five patients were over 90% $ERG-$ and over 90% $ERG+$ in two patients (Supplementary Fig. 6e), Tumor cells in four patients harbored both types of tumor cells. Using non-tumor epithelial cells as a reference, we found significantly different CNV profiles from the reference for each patient, further validating our tumor cell identification (Supplementary Fig. 6f). For our downstream analyses, we classified patients based on ERG status by annotating the five patients with almost exclusive $ERG-$ tumor cells as $ERG-$ patients and the other six patients (exclusive $ERG+$ tumor cells and mixtures) as $ERG+$ patients.

**T-cell and stromal cell analysis reveals common signaling in $ERG-$ patients.** The transcriptional differences between $ERG+$ and $ERG-$ tumor cells suggested that they might give rise to differential responses in the tumor microenvironment. To identify tumor-related immune cells and whether specific immune cell types were differentially enriched in either $ERG+$ or $ERG-$ samples, we analyzed the T-cell population and identified CD4 and CD8 T-cells, regulatory T-cells (Treg), and NK cells based on differentially expressed genes (Fig. 7a). We then stratified the T-cell populations based on $ERG$ status and found two CD4 T-cell clusters that were differentially enriched. Of two CD4 T-cell clusters we identified, CD4 T-cell cluster 1 was enriched in $ERG+$ patients with a 2.73-fold difference (20.5% vs 7.5%) (Fig. 7b) and was characterized by a higher-level expression of immune response regulators including AP-1 receptors[50] $FOSB$ (log2 fold change (log2FC) = 1.79, FDR $q$ = 5e-30), $FOS$ (log2FC = 1.78, FDR $q$ = 6.2e-26) and $JUN$ (log2FC = 1.55, FDR $q$ = 5.5e-22) (Supplementary Data 6). CD4 T-cell cluster 2 was enriched in $ERG-$ patients with a 5.6 fold difference (9.5% vs 1.7%) (Fig. 7b) ($P$ < 2.2e-16, two-sided FET) and was marked by higher expression of $CXCR6$ (log2FC = 1.31, FDR $q$ = 1.5e-22), which was previously shown to be expressed in the type-1 polarized T-cell subset and to contribute to tumor progression[51], and $DUSP4$ (log2FC = 1.30, FDR $q$ = 1.4e-20). We noted that the DEGs between the two T-cell clusters were consistent with the DEGs identified between $ERG+$ and $ERG-$ tumor cells, with $FOSB$, $FOS$, and $JUN$ overexpressed in $ERG+$ tumor cells while $CXCR6$ and $DUSP4$ were overexpressed in $ERG-$ tumor cells (Supplementary Fig. 6d). No other T-cell populations (CD8 T-cells, Treg, and NK cells) showed a significant difference in cell-type abundance between $ERG+$ and $ERG-$ patients.

Similarly, we stratified the stromal population based on the $ERG$ status of patients and identified three distinct populations, including endothelial cells, smooth muscle cells, and fibroblasts (Fig. 7c). Of these three stromal cell types, fibroblasts showed an enrichment in $ERG+$ patients ($P$ < 2.2e-16, two-sided FET) (Fig. 7d).

To test if the differences in the tumor cells between $ERG-$ and $ERG+$ patients could potentially drive distinct and common stromal and immune responses, we ran independent GSEA analysis between $ERG-$ and $ERG+$ tumor cells, CD4 T-cells, and stromal cells and computed the intersection of significantly upregulated gene sets in $ERG-$ patients (FDR $q$ < 0.1, Wilcoxon rank-sum test). Fourteen upregulated gene sets were identified that were commonly upregulated in $ERG-$ tumor cells, CD4 T-cells, and stromal cells ($P$ < 10e-20, multiset intersection exact test[52]) (Fig. 7e). However, GSEA analysis between $ERG+$ and $ERG-$ patients for tumor cells, BE, non-malignant LE, and club cells using the C2CP gene-set collection did not detect any common pathway changes shared by these epithelial cell types. BE cells in $ERG-$ patients were found to be significantly upregulated ($q$ < 0.05) in 19 pathways in all 2500 C2CP collection,

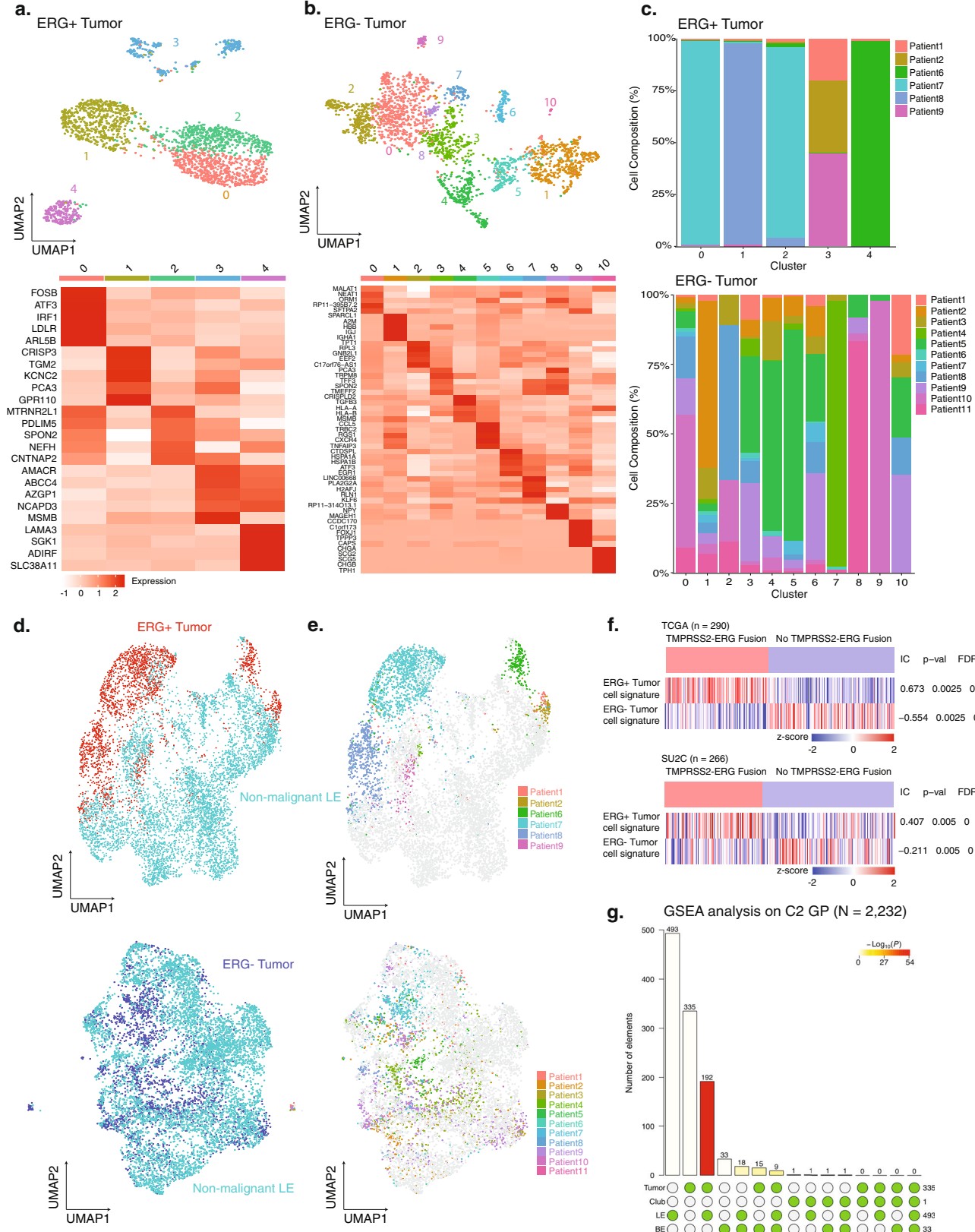

but no pathway was found to be significantly downregulated ($q < 0.05$). For club cells, no pathway was found to be significantly upregulated or downregulated in *ERG−* patients ($q < 0.05$) (Fig. 6f and Supplementary Data 5). The fourteen common upregulated gene sets in *ERG−* patients included Reactome PD-1 and Reactome interferon-gamma signaling (Fig. 7g), which have both been reported to be upregulated in advanced prostate cancers[53,54]. Within these two gene sets, we found that *ERG-* patient-enriched

**Fig. 6 Comparison of *ERG*+ and *ERG*− tumor cells reveals patient-specific cell states and intra-patient heterogeneity. a** UMAP of *ERG*+ tumor cells labeled by clusters with differential gene expression profiles (top). Heatmap of the top ten differentially expressed genes for each cluster (bottom). **b** UMAP of *ERG*− tumor cells labeled by clusters with differential gene expression profiles (top). Heatmap of the top ten differentially expressed genes for each cluster (bottom). **c** Patient composition in each cluster for *ERG*+ tumor cells (top) and *ERG*− tumor cells (bottom). Cell counts in each cluster are normalized to 100%. **d** UMAP of *ERG*+ and *ERG*− tumor cells when integrated with non-malignant LE cells, respectively. **e** UMAP of *ERG*+ and *ERG*- tumor cells when integrated with non-malignant LE cells labeled by patients. **f** The association of *TMPRSS2-ERG* fusion status in the TCGA ($N = 290$) and SU2C ($N = 266$) datasets with *ERG*+ and *ERG*− tumor cell signature (red: *TMPRSS2-ERG* fusion detected; blue: *TMPRSS2-ERG* fusion not detected). Information coefficient, accompanied *P* values and FDR *q* values are labeled. **g** Visualization of the intersection amongst significant GSEA results for BE, LE, and club cells. The color intensity of the bars represents the *P* value significance of the intersections. Source data are provided as a Source Data file.

CD4 T-cells, tumor cells, and stromal cells showed significantly higher expression of a family of HLA genes compared to *ERG*+ cell populations ($P < 0.05$, Wilcoxon rank-sum test) (Fig. 7g). Within the T-cells, while there was no difference in the cell composition of CD8 T-cells based on *ERG* status, the *ERG*− CD8 T-cell population was also found to be upregulated in the Reactome PD-1 and Reactome interferon-gamma signaling signatures (FDR $q < 0.1$, Supplementary Data 5). To test if *ERG*− tumor cell-associated CD4 and CD8 T-cells could represent a distinct immune cell niche, we tested a series of exhausted, cytotoxic markers[55] as well as genes in the PD-1 and Reactome interferon-gamma signaling pathway (Supplementary Data 7). We found that *ERG*− CD4 T-cells were significantly upregulated in exhausted T-cell markers *PDCD1* (log2FC = 0.52, $P < 0.01$, Wilcoxon rank-sum test) and *CTLA4* (log2FC = 1.79, $P < 0.001$, Wilcoxon rank-sum test) and cytotoxic markers *GZMA* (log2FC = 1.54, $P < 0.001$, Wilcoxon rank-sum test) and *GZMB* (log2FC = 1.09, $P < 0.05$, Wilcoxon rank-sum test) compared to *ERG*+ CD4 T-cells (Supplementary Fig. 7a, b). In addition, *ERG*− CD8 T-cells were upregulated in exhausted T-cell markers *HAVCR2* (log2FC = 0.68, $P < 0.05$, Wilcoxon rank-sum test) and *LAG3* (log2FC = 0.86, $P < 0.001$, Wilcoxon rank-sum test) compared to *ERG*+ CD8 T-cells (Supplementary Fig. 7a, b). These results suggested that CD4 and CD8 T-cells associated with *ERG*− tumor cells represented a more exhausted and cytotoxic phenotype. Then, using CD4 phenotype markers from a previous analysis[56], we tested the expression of these markers in both *ERG*+ and *ERG*− CD4 T-cells and found a significantly higher proportion of *CCR7*+ central memory CD4 T-cells, *GZMB*+ cytotoxic CD4 T-cells and *TOX*+ exhausted CD4 T-cells[56] associated with *ERG*− patients (Supplementary Fig. 7c). However, when testing T-cell activation markers such as *CD69, TRFC, IL2RA*, and *HLA-DRA* in both CD4 T-cells (Supplementary Fig. 7d) and CD8 T-cells (Supplementary Fig. 7e), no statistically significant difference in expression was detected between *ERG*+ and *ERG*− tumors, suggesting that *ERG* status was unlikely to be associated with higher T-cell activation.

Aside from T-cells, myeloid cells comprised the second largest immune cell population. Annotation of the myeloid cell population with SingleR[19] yielded four cell types: neutrophils, eosinophils, macrophages, and monocytes (Supplementary Data 8 and Supplementary Fig. 8a, b). Within the myeloid cells, we did not detect any significant composition differences in monocytes or macrophages between RP paired tumor and paired normal samples or between *ERG*+ and *ERG*− patients ($P > 0.05$, two-sided FET) (Supplementary Fig. 8c).

To investigate the subtypes of monocytes and macrophages that are associated with tumor-related responses, we identified monocytes and macrophages with high expression of cell cycle markers *MKI67* and *TOP2A*, indicating a cluster of proliferating myeloid cells (Supplementary Fig. 8d) that we termed *MKI67*+ myeloid cells. Monocytes were further classified by the expression of *CD14* (Supplementary Fig. 8d). Within the macrophage population, we used previously established signatures[57–60] of dichotomous phenotypes to classify macrophages into M0, M1,

and M2 types, of which M1 macrophages have been described as pro-inflammatory and M2 macrophages as anti-inflammatory and associated with tumor progression[61]. We computed the signature scores of M1 and M2 macrophages and annotated the two subtypes accordingly, based on signature scores as well as M1 specific markers such as *IL1A*, *CXCL3*, and *PTGS2*, and M2-specific markers such as *ARG1*, *CCL22*, and *FLT1*. Neither M1 nor M2 macrophages were clustered separately from normal M0 macrophages, consistent with a previous analysis of macrophage subtypes[59] (Supplementary Fig. 8d, e).

A recent study on macrophages categorized macrophages into resident tissue macrophages enriched in normal tissues (RTM) and tumor-associated macrophages (TAM) enriched in tumor tissues, which did not fit the M1/M2 phenotypes[62,63]. We did not detect RTMs within the PCa samples (Supplementary Fig. 8f). In contrast, TAMs were described as either *C1QC*+ or *SPP1*+. These TAMs were reported to derive from *FCN1*+ monocyte-like macrophages, which was consistent with the detection of *FCN1* in a cluster of PCa myeloid cells where we saw a mixture of monocytes and macrophages (Supplementary Fig. 8f). In total, 713 TAMs were identified but no significant difference in composition was detected between paired tumor and normal samples (77.9% vs 69.0%, $p = 0.58$, two-sided FET) (Supplementary Fig. 8g).

Another group of tumor-associated myeloid cells termed myeloid-derived suppressor cells (MDSC) has been characterized with roles in inflammation, establishing host immune homeostasis, and driving castration resistance in prostate cancer[64–67]. These MDSCs can inhibit anti-tumor reactivity of T cells and NK cells and the enrichment of MDSCs was correlated with tumor progression and worse clinical outcomes[68]. Two types of MDSCs have been described: monocytic MDSC (M-MDSC) characterized by high expression of *CD11* and *CD14* and low expression of *HLA* and *CD15* and granulocytic or polymorphonuclear MDSC (PMN-MDSC) characterized by high expression of *CD11* and *CD15* and low expression of *CD14*. To test for the presence of these MDSCs in our PCa samples, we used the co-expression of these markers and identified 137 M-MDSCs within the 790 *CD14*+ monocytes and 11 PMN-MDSCs within 974 *CD14*- monocytes (Supplementary Fig. 8g). M-MDSCs were enriched in the paired tumor samples compared to paired normal (19.9% vs 3.6% of total monocytes, $P = 0.0035$, two-sided FET).

**Prostate cancer organoids harbor epithelial cell types with uniquely expanded cell states in BE and club cells.** To develop models to examine the cellular state heterogeneity revealed by single-cell analysis and to determine if we could reconstitute and propagate prostate cancer-associated club cells, we used established methods[69,70] to generate localized prostate cancer organoids from single cells from six patients who underwent radical prostatectomies (four patients included in the tissue sample dataset) and characterized them using scRNA-seq within three passages (Fig. 8a). PCA-based clustering of organoid samples

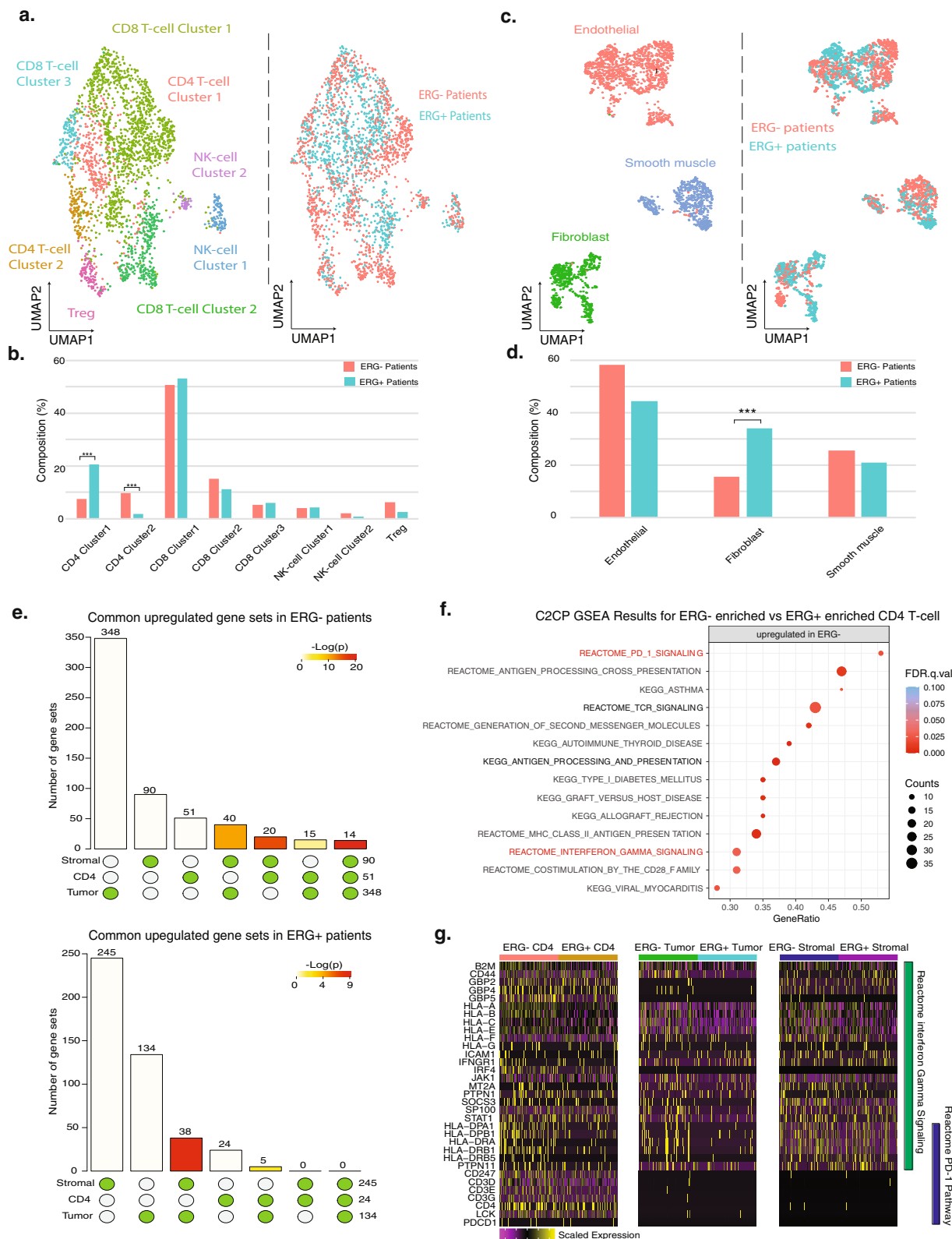

yielded 23 clusters from a total of 15,073 cells. We identified a total of six epithelial cell types with distinctive DEGs, based on cell-type signatures we generated from PCa tissue samples and established signatures from normal samples (Supplementary Data 2) (Fig. 8a), including BE cells characterized by high expression of *DST, KRT15, KRT5, KRT17,* and *TP63*, club cells characterized *by PIGR, MMP7, CP,* and *CEACAM6*, hillock cells,

consistent with those in normal prostates showing high-level expression of *KRT13, CLCA4,* and *SERPINB3*, a mesenchymal stem cell (MSC) population expressing known MSC markers[71–73] *LAMC2, VIM, MMP1,* and *KLK7* and a population with high-level expression of cell cycle markers *MKI67* and *TOP2A* termed *MKI67* + epithelial cells (Supplementary Fig. 9a). Notably, within these early-passage organoids we identified a population fitting

**Fig. 7 CD4 T-cell subsets associated with *ERG* status and common upregulation of PD-1 and interferon-gamma signaling in the *ERG*− tumor microenvironment. a** UMAP of T-cells labeled by different cell types (left) and *ERG+* or *ERG*− patients (right). **b** Cell composition comparison between *ERG+* and *ERG*− patients for all T-cell cell types. Significance levels are labeled in differentially enriched clusters (***P < 0.001, two-sided FET; CD4 T-cell cluster 1: *P* = 4.28e-13; CD4 T-cell cluster 2: *P* = 1.08e-27). **c** UMAP of stromal cells labeled by different cell types (left) and *ERG+* or *ERG*− patients (right). **d** Cell composition comparison between *ERG+* and *ERG*− patients for all stromal cell types (***P < 0.001; *P* = 1.15e-31, two-sided FET). Significance levels are labeled in differentially enriched clusters. **e** Visualization of the intersections amongst significantly upregulated (top) and downregulated (bottom) gene sets within C2CP gene-set collection for tumor cells, two clusters of differentially enriched CD4 T-cell clusters, and stromal cells. Significant Gene Set Enrichment Analysis (GSEA) results are represented by circle below bar chart with individual blocks showing "presence" (green) or "absence" (gray) of the gene sets in each intersection. *P* value significance of the intersections are represented by the color intensity of the bars. **f** GSEA results for the *ERG*- patient-enriched CD4 T-cell cluster compared to the *ERG+* patient-enriched cluster on the common upregulated gene sets (*N* = 14). Gene counts for the corresponding gene set are indicated by marker radius. Statistical significance levels (FDR) are shown by color gradient. Reactome PD-1 and Interferon-gamma signaling pathways are highlighted in red. **g** Gene expression heatmaps of genes in the Reactome PD-1 and Interferon-gamma signaling pathways for tumor cells, CD4 T-cells and stromal cells in both *ERG+* and *ERG*− patients. Source data are provided as a Source Data file.

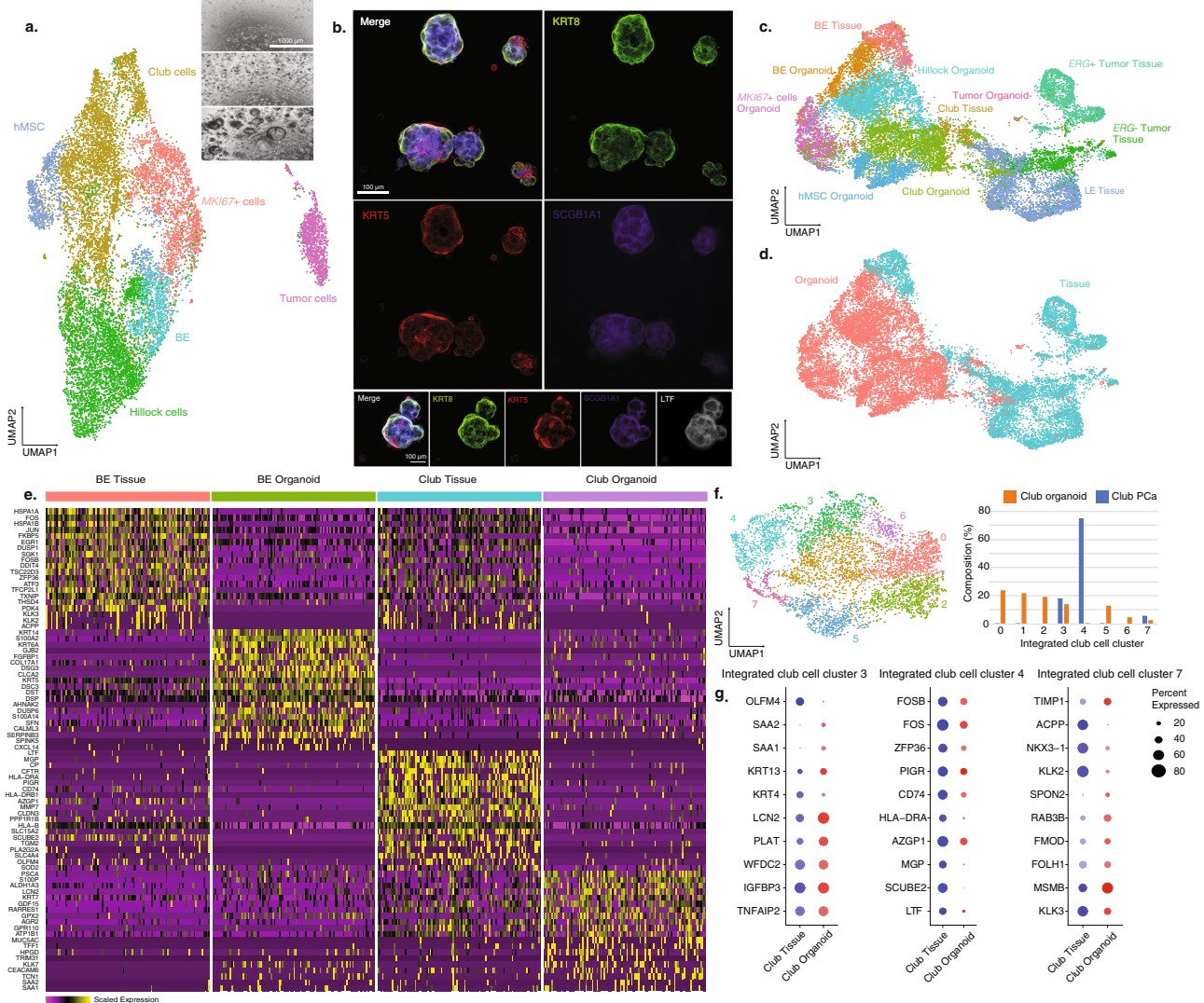

**Fig. 8 In vitro organoid samples harbor PCa-enriched BE and club cell states. a** UMAP of cells from organoid samples labeled by different cell types. Organoid culture snapshots are depicted in the upper right panel. **b** Immunofluorescence staining for LE marker (*KRT8*), BE marker (*KRT5*) and club cell markers (*SCGB1A1, LTF*) of the organoid samples. Technical and biological replicates for four additional organoids reproduce the shown staining. Due to the limitation of organoid sizes and subsequently image quality, only one group of experimental results is shown. **c** UMAP of the integrated dataset of cells from the organoid samples and epithelial cells from matching parent tissue samples, labeled by cell types. **d** UMAP of the integrated dataset, labeled by sample types (tissue or organoid samples). **e** Heatmaps for the top 20 differentially expressed genes for BE and club cells between tumor tissues and organoid samples. **f** UMAP of integrated club cell dataset of tumor tissue and organoid samples. Cell composition comparison is shown in the grouped bar charts. **g** Dot plots of the top ten differentially expressed genes in clusters 3, 4, and 7 in tissue and organoid club cells. Dot size represents proportions of gene expression in cells and expression levels are shown by color shading (low to high reflected as light to high). Source data are provided as a Source Data file.

our profiling of ERG+ tumor cells, expressing a high level of LE cell markers (*KLK3, KLK2,* and *ACPP*) and tumor markers (*PCA3, TRPM8,* and *ERG*) (Supplementary Fig. 9a), which we annotated as putative tumor cells. Cell-type annotation was supported by ssGSEA, which showed that the MSC population was upregulated in the MSC signature gene set developed from a previous analysis[72] and that the *MKI67+* cluster was upregulated in a KEGG cell cycle signature indicating proliferating cells (Supplementary Fig. 9b). The tumor cells were validated by InferCNV[21] estimation (Supplementary Fig. 9c). To confirm our recovery of the cell-type diversity in the organoids, we performed immunofluorescence staining for *KRT8+* luminal and *KRT5+* basal cells (Fig. 8b). We validated club cell proliferation in vitro by staining for *SCGB1A1*, an established club cell marker in the lung and prostate[9], and lactoferrin (*LTF*), which was upregulated in the PCa-club cells identified by scRNA-seq (Fig. 8b).

To test the fidelity of the organoids as models for tumor tissues, we integrated the cells in the early-passage (P0-P3) organoid samples (N = 10,990) with the epithelial cells from the four RP specimens from which the organoids were derived (N = 8719) (Fig. 8c). Compared to PCa tissue samples, in the organoid samples, LE cell markers or signature scores could not identify a distinctive LE cell cluster in the organoid samples (Supplementary Fig. 9b), consistent with a previous study that LE cells were rarely captured in in vitro organoid cultures analyzed by scRNA-seq[74]. For the four patient-derived organoids, only a small number of tumor cells were captured compared to the parent tissues (tissue samples vs organoids, 34.11% vs 0.11%). However, hillock cells, MSCs, and a population of *MKI67+* epithelial cells were exclusive to the organoid samples and were not observed in PCa tissue samples (Fig. 8d).

As BE and club cells were the two primary overlapping cell types between tissue and organoid samples (representing 11.9% and 29.0% of all cells, respectively, in the organoid samples), we took the subset of BE cells and club cells in tissue and organoid samples from the integrated dataset and computed the DEGs. BE markers *KRT5, DST,* and *KRT15* were expressed in BE populations from tissue and organoids and club cell genes *MMP7, LCN2,* and *CP* were expressed in both club cell populations (Fig. 8e), suggesting similarities between tissue and organoid BE and club cells.

We then investigated BE and club cell populations by integrating organoids with tissue samples, respectively, to identify cell state differences in the organoid samples. We identified nine clusters in the integrated BE cell dataset with distinctive groups of DEGs (Supplementary Fig. 9d). Compared to BE cells in PCa tissue samples, significantly higher percentages of BE cells in organoids expressed *KRT6A* (organoid vs tissue, 77.4% vs 0.56%, P < 2.2e-16, two-sided FET), *KRT14* (organoid vs tissue, 71.2% vs 18.6%, P < 2.2e-16, two-sided FET) and *KRT23* (organoid vs tissue, 78.8% vs 20.2%, P < 0.001, two-sided FET) (Supplementary Fig. 9e).

Similarly, when analyzing the organoid club cells with club cells from PCa tissues, we identified a total of eight clusters with distinctive DEGs (Supplementary Fig. 9f) and observed an expansion of cell states in the organoid samples (Fig. 8f). Among the eight clusters, five were predominantly comprised of organoid club cells while club cells from prostate tissue were only found in clusters 3, 4, and 7. By comparing the expression levels of the top differentially expressed genes for these three clusters split by tissue and organoid club cells, we found that in cluster 3, hillock cell marker *KRT13* was expressed in tissue and organoid club cells, suggesting an intermediate hillock-club cell state. In cluster 4, PCa-club cell marker *PIGR* was detected in 47% of organoid club cells (16 of 34) and 71% of tissue club cells (325 of 653). *LTF* was expressed in 15% of organoid club cells (5 of 34) compared to

50% tissue club cells (326 of 653), suggesting that *LTF* may be a PCa tissue-specific club cell marker. In contrast, the top DEGs for cluster 7 included LE markers such as *ACPP, NKX3-1, KLK2,* and *KLK3*, consistent with the profile of the previously identified PCa-enriched club cell state (Fig. 8g). In cluster 7, we observed approximately 20% of organoid club cells expressing at least one LE cell marker. This cluster scored higher for the PCa-enriched club cell state compared to all other clusters of organoid club cells, suggesting that PCa-enriched club cell states were also observed in organoid samples. Overall, we found that organoid samples harbored cell states found in tumor tissues and an enrichment of intermediate cell states. These organoid-enriched cell states within BE and club cells suggest that in vitro organoid models may provide useful models to study cell-state differences and identify lineage relationships to tumorigenesis.

## Discussion

Studies of localized prostate cancer have been extensively performed with bulk RNA-seq and WES/WGS approaches that have provided key insights into the molecular features of prostate cancer[9,12,63–66,75–77]. Here, we performed single-cell analyses of localized PCa biopsies and radical prostatectomy specimens to characterize the heterogeneity of tumor cells, subpopulations of epithelial cells, stromal cells, and tumor microenvironments.

Of note, we identified a distinctive epithelial cell population of club cells that has not been previously observed in human prostate cancer samples. While club cells have been noted in normal prostates[9,78,79], a population of club cells associated with prostate cancer suggests they may play a previously unappreciated role in carcinogenesis. Recent studies have identified a progenitor-like $CD38^{low}$ $PIGR^{high}$ $PSCA^{high}$ luminal epithelial cell sub-population with regenerative potential[39,45,79]. Based on the similarity of highly expressed genes including *PIGR, MMP7, CP,* and *LTF*, we believe those cells are consistent with their identity as club cells. In our analysis, prostate cancer club cells are characterized by the markedly lower expression of *SCGB3A1* and *LCN2* compared to club cells from normal healthy controls[9]. Based on our gene signature analyses, our results suggest that PCa-club cells are more androgen-responsive overall and harbor a highly androgen-responsive cell state that may play a supporting role for the overall androgen-responsive cellular milieu of prostate cancer[80,81].

*SCGB3A1*, a marker for club cells, was one of the top downregulated genes in prostate cancer club cells compared to club cells from normal healthy control prostates. *SCGB3A1* may play a tumor suppressor role in a number of cancers including breast, prostate, and lung as its expression has been noted to be markedly lower in cancer tissues compared to normal tissue[82]. Prostate club cells in the normal epithelia may play a tumor suppressor role through secretion of *SCGB3A1* which is then downregulated in concert with prostate cancer progression, as marked by our finding of $SCGB3A1^{low}$ club cells in prostate cancer tissues that can be propagated in organoids. We did not find a distinct population of hillock cells in prostate cancer tissues so it is possible that hillock cells may be depleted in prostate cancer progression.

Consistent with other cancer single-cell studies in which tumor cells cluster separately, *ERG+* tumor cells clustered separately by patient from non-malignant epithelial clusters[14,55,83–85]. However, our analysis of *ERG−* tumor cells unexpectedly found that *ERG−* tumors did not cluster by patient and we observed a shared heterogeneity for *ERG-* tumor cells with non-malignant luminal cells. The identification of tumor cells in this study was largely based on the CNV estimation using the non-malignant epithelial cells as reference. Given the relatively shallow sequencing depth of

scRNA-seq, we acknowledge that it was possible that small focal CNV changes might not be well captured in our analysis.

Our single-cell analysis reveals insights into the tumor immune microenvironment of localized prostate cancer based on *ERG* status. We hypothesized that *ERG−* tumor cells might evoke similar tumor microenvironment responses and found common transcriptional pathways that were upregulated in the tumor, stroma, and CD4 T-cell populations of *ERG−* patients, including the PD-1 and interferon-gamma signaling pathway, suggesting that *ERG-* tumor cells may give rise to a distinct immune cell niche and tumor microenvironment. While immune checkpoint inhibition (ICI) has had limited efficacy in advanced castration-resistant prostate cancers[42,44,86–90], we speculate that there may be targeted trials in *ERG-* hormone-sensitive prostate cancers that may be more beneficial with ICI or other emerging immunotherapies that are currently in clinical trials.

We note a potential limitation of our analysis in the identification of *ERG−* tumor cells as we also found evidence for *ERG−* tumor cells in paired grossly normal specimens. This could be attributed to tumor cells also being present in the seemingly normal tissues from radical prostatectomy specimens[14,85,91,92]. Analysis of somatic mutations or structural variants on a single-cell level will help confirm our identification of *ERG-* tumor cells and inform our understanding of their heterogeneity.

Furthermore, we found that PCa-enriched epithelial cell states identified in the tumor tissues were also present in in vitro organoid cultures grown from tumor specimens. We identified cell types that emerged in the organoid samples but were not observed in the tumor tissues, including hillock cells, MSC, and *MKI67+* epithelial cells. The mechanisms by which hillock cells can propagate in organoid cultures but not be found in the localized tumor tissue specimens are still to be delineated. An expansion of cell states in BE and club cells in the organoids suggests a broader view of their capacity for cell-state transitions. Our analysis suggests that prostate cancer epithelial organoids harbor many major cell types from tissue and may provide a useful model to investigate cell-state plasticity in the context of selective pressures and genetic perturbations. However, in contrast to previous studies on organoids generated from prostate samples, we did not observe a distinctive *NKX3-1+/KLK3+/AR+* luminal cell population[69,93,94]. This might be due to a limitation of detection using single-cell sequencing technology or that we could not robustly grow differentiated luminal cells[74]. We detected a population of "tumor-like" cells largely from one patient organoid. It is possible that these "tumor-like" cells are present only initially in some organoids but dissipate after several passages.

Comparing epithelial cells from PCa samples with those from normal healthy controls revealed distinct high androgen-signaling cell states that were enriched in PCa samples. We found that epithelial cells from PCa tissues were generally upregulated in *AR* signaling. Given our identification of shared luminal-like, highly androgen-responsive cell states across basal and club cell populations, we posit that these cell types may be primed for tumor cell transformation and may also promote prostate tumorigenesis. Further studies with lineage tracing and dissection of single-cell somatic alterations within these specific cell states will be informative for further characterization of their potential tumorigenic roles. The identification of a tumor-associated club cell population raises the possibility that these cells contribute to the interactions between tumor cells and their surrounding epithelial microenvironment. Furthermore, our analyses identify cell-type-specific signature gene sets within prostate cancer samples that should contribute to a more precise and thorough classification of cells during prostate carcinogenesis. In summary, we provide a single-cell transcriptomic blueprint of localized prostate cancer that identifies and highlights the multicellular milieu and cellular states associated with prostate tumorigenesis. Our results provide insights into the epithelial microenvironment and the cellular state changes associated with prostate cancer toward improved PCa diagnosis.

## Methods
### Experimental details
*Sample collection.* We obtained a total of six prostate biopsies from three different patients (two biopsies each for patient 1–3, obtained at the same time point), four radical prostatectomies (RP) with tumor-only samples from four patients (patients 4–7), and four radical prostatectomies with matched normal samples from four patients (patients 8–11). All RP patients had lesions that were visible on preoperative MRI or ultrasound and were later confirmed pathologically to be cancer (Supplementary Fig. 2a). Matched normal samples were taken from normal regions). Clinical/pathological data available for the samples is in Supplementary Data 1. Of these 11 patients, only one patient (patient 4) was treated with finasteride.

*Study approval.* The UCSF Institutional Review Board (IRB) committee approved the collection of these patient data included in this study. All relevant ethical regulations for work with human participants have been compiled and written informed consent was obtained.

*Tissue dissociation.* Tissue samples were minced with surgical scissors and washed with RP-10 (RPMI + 10% FBS). Each sample was centrifuged at 259×g for 5 min, resuspended in 10 mL digestive media (HBSS + 1% HEPES) with Liberase TM (Roche, Cat: 5401119001) or 1000 U/mL collagenase type IV (Worthington, Cat: LS004188), and rotated for 30 min at 37 °C. Samples were triturated by pipetting ten times after every 10 min during the incubation or by pipetting 15 times at the end of the incubation. Each sample was filtered through a 70-μm filter (Falcon, Cat: 352350), washed with RP-10, centrifuged at 259×g for 5 min, washed again with RP-10, and resuspended in RP-10. A hemocytometer was used to count the cells.

*Single-cell RNA sequencing.* Sequencing was largely based on the Seq-Well S^3 protocol[13,95]. One to four arrays were used per sample. Each array was loaded as previously described with 110,000 barcoded mRNA capture beads (ChemGenes, Cat: MACOSKO-2011-10(V+)) and with 10,000–20,000 cells. Arrays were sealed with functionalized polycarbonate membranes (Sterlitech, Cat: PCT00162X22100) and were incubated at 37 °C for 40 min.

After sealing, each array was incubated in lysis buffer (5 M guanidine thiocyanate, 1 mM EDTA, 0.5% sarkosyl, 1% BME). After detachment and removal of the top slides, arrays were rotated at 50 rpm for 20 min. Each array was washed with hybridization buffer (2 M NaCl, 4% PEG8000) and was then rocked in a hybridization buffer for 40 min. Beads from different arrays were collected separately. Each array was washed ten times with wash buffer (2 M NaCl, 3 mM MgCl₂, 20 mM Tris-HCl pH 8.0, 4% PEG8000) and scraped ten times with a glass slide to collect beads into a conical tube.

For each array, beads were washed with Maxima RT buffer (ThermoFisher, Cat: EP0753) and resuspended in reverse transcription mastermix with Maxima RT buffer, PEG8000, Template Switch Oligo, dNTPs (NEB, Cat: N0447L), RNase inhibitor (Life Technologies, Cat: AM2696) and Maxima H Minus Reverse Transcriptase (ThermoFisher, Cat: EP0753) in water. Samples were rotated end-to-end, first at room temperature for 15 min and then at 52 °C overnight. Beads were washed once with TE-SDS and twice with TE-TW. They were treated with exonuclease I (NEB), rotating for 50 min at 37 °C. Beads were washed once with TE-SDS and twice with TE-TW, and once with 10 mM Tris-HCl pH 8.0. They were resuspended in 0.1 M NaOH and rotated for 5 min at room temperature. They were subsequently washed with TE-TW and TE. They were taken through second strand synthesis with Maxima RT buffer, PEG8000, dNTPs, dN-SMRT oligo, and Klenow Exo- (NEB, Cat: M0212L) in water. After rotating at 37 °C for 1 h, beads were washed twice with TE-TW, once with TE and once with water.

KAPA HiFi Hotstart Readymix PCR Kit (Kapa Biosystems, Cat: KK2602) and SMART PCR Primer (Supplementary Data 9) were used in whole transcriptome amplification (WTA). For each array, beads were distributed among 24 PCR reactions. Following WTA, three pools of eight reactions were made and were then purified using SPRI beads (Beckman Coulter), first at 0.6× and then at a 0.8× volumetric ratio.

For each sample, one pool was run on an HSD5000 tape (Agilent, Cat: 5067–5592). The concentration of DNA for each of the three pools was measured via the Qubit dsDNA HS Assay kit (ThermoFisher, Cat: Q33230). Libraries were prepared for each pool, using 800–1000 pg of DNA and the Nextera XT DNA Library Preparation Kit. They were dual-indexed with N700 and N500 oligonucleotides.

Library products were purified using SPRI beads, first at 0.6× and then at a 1× volumetric ratio. Libraries were then run on an HSD1000 tape (Agilent, Cat: 50675584) to determine the concentration between 100–1000 bp. For each library, 3 nM dilutions were prepared. These dilutions were pooled for sequencing on a NovaSeq S4 flow cell.

The sequenced data were preprocessed and aligned using the dropseq_workflow on Terra (app.terra.bio). A digital gene expression matrix was generated for each sample, parsed, and analyzed following a customized pipeline. Additional details are provided below.

*Whole-exome sequencing.* The remaining frozen single cells from the prostate tumor and matching normal specimens ($N = 3$ patients) were processed for genomic DNA and underwent whole-exome sequencing (Novogene). Sequencing data were then aligned to the GRCh38/hg38 reference genome. Somatic single-nucleotide variants (SNVs) were identified using our in-house pipeline which integrated somatic variant caller Mutect2 and annotation using Funcotator[96]. The list of SNVs was further filtered using the following criteria: (a) variants with less than a minimum read depth of ten reads were excluded, (b) variants with less than three supporting reads of the altered nucleotide were excluded, (c) variants with a variant allele frequency of less than 5% were excluded. Somatic copy number alterations were identified using the GATK somatic CNV pipeline[96].

*Organoid culture.* Isolated single cells not used for single-cell sequencing were additionally frozen in FBS + 10% DMSO, flash-frozen on dry ice, or plated in Matrigel to grow as 3D prostate organoid cultures. Organoid cultures were established by plating 20,000 cells in 25 µL Matrigel (Corning, Cat: 356231) in 48-well flat-bottom plates (Corning, Cat: EK-47102). Prostate-specific serum-free culture media contained 500 ng/mL human recombinant R-spondin (R&D Systems, Cat: 10820-904), 10 µM SB202190 (Sigma, Cat: S7076), 1 µM prostaglandin E3 (Tocris, Cat: 229610), 1 nM FGF10 (Peprotech, Cat: 100-26), 5 ng/mL FGF2 (Peprotech, Cat: 100-18B), 10 ng/mL 5 alpha-dihydrotestosterone (Sigma, Cat: D-073-1ML), 100 ng/mL human Noggin (Peprotech, Cat: 102-10 C), 500 nM A83-01 (Fischer, Cat: 29-391-0), 5 ng/mL human EGF (Peprotech, Cat: AF-100-15), 1.25 mM N-acetyl-cysteine (Sigma, Cat: A9165), 10 mM Nicotinamide (Sigma, Cat: N3376), 1× B-27 (Gibco, Cat: 17504044), 1× P/S (Gibco, Cat: 15140122), 10 mM HEPES (Gibco, Cat: 15630080), and 2 mM GlutaMAX (Gibco, Cat: 35050061)[70]. In addition, 10 µM Y-27 (Biogems, Cat: 1293823) was included during the first 2 weeks of growth and after passaging to promote growth[70]. Generally, organoid growth was apparent within 2–3 days and robust after 2 weeks. In total, 250 µL media was refreshed every 2–4 days using media stored at 4 °C for a maximum of 10 days. Organoid growth was monitored using an EVOS-FL microscope.

To passage prostate organoid cultures every 7–14 days, culture media was replaced with 300 µL TrypLE (1×, Gibco, Cat: 12604013). Individual domes were collected into 15-mL Falcon tubes, disrupted by pipetting with wide-orifice tips, and incubated at 37 °C for 30 min. Following incubation, the dissociation media was neutralized using 10 mL wash media: adDMEM/F12 containing 5% FBS, P/S, 10 mM HEPES (1 M, Gibco, Cat: 15630080) and 2 mM GlutaMAX (100×, Gibco, Cat: 35050061)[70]. Cells were spun down at 500 G for 5 min and resuspended in 2 mL wash media. Finally, the media was aspirated, cells were resuspended in Matrigel, and 25 µL/dome were plated per well.

Organoids were accessed using single-cell sequencing at an early passage (P0-4). To isolate single cells from Matrigel, organoids were collected in 500 µL Trypsin (0.25%, Gibco, Cat: 25-200-056) and incubated at 37 °C for 30–45 min until few clumps were visible. Throughout incubation, cells were triturated every 5 min. Single cells were resuspended in 9 mL DMEM + 5% FBS + 0.05 mM EDTA and passed through a 40-µm filter, followed by an additional wash of the filter with 1 mL DMEM + 5% FBS + 0.05 mM EDTA. Cells were spun down at 300 G for 5 min, resuspended in 10 mL of the same media, spun down again, and finally, resuspended in 1–2 mL media. Cells were counted using a hemocytometer and loaded onto arrays for single-cell sequencing as described for patient tissues.

*Immunofluorescence.* Organoids were passaged into eight-well Nunc Lab-Tek II Chamber Slides (Thermo Scientific, Cat: 154453) and allowed to grow in prostate-specific media. Following 7 days, the media was removed, domes were washed twice with 300 µL PBS and fixed in 4% paraformaldehyde (Electron Microscopy Sciences, Cat: 15710-S) at room temperature for 20 min. Individual domes were washed 3× with IF Buffer (0.02% Triton + 0.05% Tween + PBS) and blocked for 1 h at room temperature with 0.5% Triton X100 + 1% DMSO + 1% BSA + 5% donkey serum + PBS. Following the block, domes were washed once with IF Buffer and counterstained with monoclonal mouse anti-Lactoferrin (Abcam, Cat: ab10110, 1 µg/mL), monoclonal rat anti-Uteroglobin/SCGB1A1 (R&D Systems, Cat: MAB4218-SP, 1 µg/mL), polyclonal guinea pig anti-Cytokeratin 8 + 18 (Fitzgerald, Cat: 20R-CP004, 1:100), and polyclonal chicken anti-keratin 5 (Biolegend, Cat: 905901, 1:100). Subsequently, domes were washed 3× with IF Buffer and counterstained with Alexa Fluor 488-AffiniPure donkey anti-chicken IgY (IgG) (H + L) (Jackson ImmunoResearch, Cat: 703-545-155, 1:500), donkey anti-mouse IgG (H + L) cross-adsorbed secondary antibody, DyLight 550 (ThermoFisher Scientific, Cat: SA5-10167, 1:500), donkey anti-rat IgG (H + L) cross-adsorbed secondary antibody, DyLight 680 (ThermoFisher Scientific, Cat: SA5-10030, 1:500), and Alexa Fluor 790 AffiniPure donkey anti-guinea pig IgG (H + L) (Jackson ImmunoResearch, Cat: 706-655-148, 1:500) containing DAPI (Sigma, Cat: D9542-5MG, 1:1000). Finally, wells were washed 3× with IF Buffer for 5 min and sealed with Prolong Gold antifade mountant (Fischer Sci, Cat: P36930). Z-stack images were captured on a Leica DCF9000 GT using Leica Application System X software.

## Quantification and statistical analysis
*Sequencing and alignment.* Sequencing results were returned as paired FASTQ reads and processed with FastQC[97] (v0.11.9) for general quality checks in order to further improve our experimental protocol. Then, the paired FASTQ files were aligned against the reference genome using a STAR aligner[98] (v2.7.6a) built within

the dropseq workflow (Snapshot 7) (https://cumulus.readthedocs.io/en/latest/drop_seq.html). The aligning pipeline output included aligned and corrected bam files, two digital gene expression (DGE) matrix text files (a raw read count matrix and a UMI-collapsed read count matrix where multiple reads that matched the same UMI would be collapsed into one single UMI count) and text-file reports of basic sample qualities such as the number of beads used in the sequencing run, total number of reads, alignment logs. For each sample, the average number of reads was 4,875,9687, and the mean read depth per barcode was 48,586. The median and the average number of genes per barcode were 767 and 1079. The median and average number of UMI were 1335 and 2447. The mean percentage of mitochondrial content per cell was 13.65%.

*Single-cell clustering analysis.* Cells in the samples were clustered and analyzed using customized codes based on the Seurat package (v3.2.2) in R[20] (v4.0.3). Cells with less than 300 genes, 500 transcripts, or a mitochondrial level of 20% or greater, were filtered out as the first QC process. Then, by examining the distribution histogram of the number of genes per cell in each sample, we set the upper threshold for the number of genes per cell in each individual sample in order to filter potential doublets. A total of 22,037 cells were acquired using these thresholds. Since merging with and without integration of the samples showed no major difference in the clustering of each cell type, in the subsequent analysis of these samples we used the merged dataset without integration.

Doublets were removed by two steps: first, we used DoubletFinder[99] (v2.0.3) and a theoretical doublet rate of 5% to locate doublets in our dataset. 305 cells marked by DoubletFinder as true positive were removed from further analysis. 21,743 cells were used in the following cell clustering analysis. Then, after clustering, we removed cells expressing biomarkers from more than one major cell type (epithelial, stromal, and immune) as they were more likely to be doublets. In this step, we removed 276 cells from our dataset and the follow-up analysis, leaving 21,467 cells in total.

UMI-collapsed read counts matrices for each cell were loaded in Seurat for analysis[20]. We followed the standard workflow by using the "LogNormalize" method that normalized the gene expression for each cell by the total expression, multiplying by a scale factor 10,000 and log-transforming the results. For downstream analysis to identify different cell types, we then calculated and returned the top 2000 most variably expressed genes among the cells before applying a linear scaling by shifting the expression of each gene in the dataset so that the mean expression across cells was 0 and the variance was 1. This way, the gene expression level could be comparable among different cells and genes. PCA was run using the previously determined most variably expressed genes for linear dimensional reduction and the first 100 principal components (PCs) were stored which accounted for 25.42% of the total variance. To determine how many PCs to use for the clustering, a JackStraw resampling method was implemented by permutation on a subset of data (1% by default) and rerunning PCA for a total of 100 replications to select the statistically significant principle component to include for the K-nearest neighbors clustering. For graph-based clustering, the first 100 PC and a resolution of 3 were selected yielding a total of 46 cell clusters. We eliminated the clustering side effect due to overclustering by constructing a cluster tree of the average expression profile in each cluster and merging clusters together based on their positions in the cluster tree. As a result, we ensured that each cluster would have at least 10 unique differentially expressed genes (DEGs). Differentially expressed genes in each cluster were identified using the FindAllMarker() function built within Seurat package and a corresponding p-value was given by the Wilcoxon's test followed by a Bonferroni correction. Top differentially expressed gene markers were illustrated in a stacked violin plot using a customized auxiliary function. Dot plots were generated as an alternative way of visualization using the top ten differentially expressed genes in each cluster. Top tier cell-type clustering was also validated by the automated singleR (v1.2.4) annotation (Supplementary Data 1). However, when running singleR in single-cell mode for epithelial cells, and due to the lack of detailed reference in the singleR library, singleR could not identify detailed epithelial cell types. Therefore, manual annotation was required for epithelial cells.

*Cell-type annotation by signature scores.* In order to annotate each cell type from the previous clustering, we took the established studies and the signatures for each cell type (Supplementary Data 2). Treating the signature score of each cell type as a pseudogene, we evaluated the signature score for each cell in our dataset using the AddModuleScore() function built within Seurat[20]. Each cluster in our dataset was assigned with an annotation of its cell type by top signature scores within the cluster.

*Epithelial sub-clustering analysis and tumor cell inference.* All epithelial cells were clustered using the analytical workflow described above, yielding 20 clusters. To compare the transcriptomic profiles between PCa samples and normal prostates, a previous study on normal prostate single-cell RNA-seq was downloaded and imported. Mean basal, luminal, hillock, and club signature scores were calculated for each cluster, based on the top differentially expressed genes from a previous scRNA-seq study on the normal prostate. A One-way ANOVA test was then conducted to determine if the signature score of each cluster was significantly different from the rest. We annotated the clusters with significantly upregulated

basal epithelial cell (BE) signature scores as BE. Cells in clusters with high luminal epithelial (LE) signature scores could be either non-malignant luminal epithelial cells or tumor cells. The clusters with low signature scores of both BE and LE were annotated as other epithelial cells (OE). To efficiently identify tumor cells, we took the digital gene expression matrix and conducted a single-set gene-set enrichment analysis on GenePattern (https://gsea-msigdb.github.io/ssGSEA-gpmodule/v10/index.html) testing against the C2 gene-set collection. Under the notion that tumor cells should have higher expression of one or more tumor markers overlapping existing prostate cancer gene sets, we projected the signatures of these prostate cancer gene sets onto our epithelial clusters and annotated tumor cell clusters as the clusters with significantly higher signature scores of at least one prostate cancer gene sets.

Approximately ~50% of prostate cancer cells from men of European ancestry harbor *TMPRSS2-ERG* fusion events, indicating high gene expression of *ERG*[100,101]. Therefore, we hypothesized a high signature score of SETLUR PROSTATE CANCER TMPRSS2-ERG FUSION UP gene set[26] would be a strong indicator of *ERG*+ tumor cells. All the other tumor cell clusters were then annotated as *ERG*- tumor cell clusters as they showed little to no *ERG* gene expression. All of the epithelial clusters with high luminal signature scores and high expression of luminal markers such as *KLK3, KLK2, ACPP, KRT8*, and *KRT18* were annotated as non-malignant luminal epithelial cells (non-malignant LE). Compared to non-malignant cells, tumor cells harbor more single-nucleotide variants and copy number variants, leading to distinctive patterns. To validate our tumor cell annotation, we ran InferCNV (v1.4.0) on *ERG*+ and *ERG*- tumor clusters with non-malignant LEs as reference[29] for estimation of copy number alterations. We classified tumor cells based on *ERG* gene expression. Then we defined patients harboring *ERG* + tumor cells as *ERG* + patients and the other patients as *ERG*- patients. This way, we were able to classify all the other cells based on the *ERG* status (epithelial, stromal, and immune cells) as either *ERG*+ or *ERG*-.

To determine if common functional changes were present in more than one cell type, we conducted gene-set enrichment analysis (GSEA) for each cell type first and imported the significantly changed gene sets to take the intersections. The statistical significance of multiset intersection was evaluated and visualized using the SuperExacTest package[52] (v1.0.7).

*Cell-state analysis.* Gene expression profile differences in epithelial cells between PCa sample and normal prostate samples were identified by integrating our PCa dataset with an established dataset on normal prostates[9]. We utilized the integration method based on commonly expressed anchor genes by following the Seurat integration vignette[20] (v3.2.2) in order to remove batch effects of samples sequenced with different technologies and possible artifacts so that the cells were comparable.

In order to better characterize the transcriptomic profile and transition of cell states among identified epithelial cells, both the tumor and paired normal samples were integrated together and separately with the epithelial cells from a normal prostate scRNA-seq dataset[9] for *KRT5*+ and *KRT15*+ basal epithelial (BE), *KLK3*+, and *ACPP*+ luminal epithelial (LE) and *PIGR*+ and *MMP7*+ club cell population together and separately. An optimal resolution value was tested using the Clustree[102] package (v0.4.3). Heatmaps of DEGs were generated to validate the cell-state differentiation. Compositions for each cell state was computed and compared between PCa samples and normal samples using Fisher's exact test.

To assess the functional roles of the PCa-enriched cell states identified within the integrated dataset, we ran GSEA analysis between the PCa-enriched cell state and all the other cell states as a whole. The top 20 downregulated and upregulated gene sets were visualized in terms of gene counts and ratio for each gene set. Using the DEGs from each cell state, we generated signature gene sets for all the cell states in BE, LE, and club cells. To validate the functional implications for the PCa-enriched cell states, we conducted a single-set gene-set enrichment analysis (ssGSEA) on PCa BE and club cells to compute the signature scores of the upregulated gene sets using the ssGSEA module on GenePattern (https://gsea-msigdb.github.io/ssGSEA-gpmodule/v10/index.html). Then, we computed the information coefficient (IC) and corresponding *P* values followed by FDR correction to evaluate the correlation between these gene sets and cell states.

*Pseudotime analysis.* To evaluate the epithelial cell states with respect to their order in the differentiation trajectory, we conducted pseudotime analysis on all epithelial and tumor cells identified in the PCa samples. We first calculated a PAGA (partition-based graph abstraction) graph using SCANPY's sc.tl.paga() function[103] (v1.8.1) and then used sc.tl.draw_graph() to generate the PAGA initialized single-cell embedding of the cell types. The diffusion pseudotime for each cell was calculated using SCANPY's sc.tl.diffmap() and sc.tl.dpt() with the root cluster as the BE cluster and then was plotted on the PAGA initialized embedding. We then visualized the gene marker changes along the pseudotime by cell type using sc.pl.paga_path().

*scRNA-seq fusion detection.* Fusion transcripts were detected using STAR-Fusion[27] (v1.6.0). STAR-Fusion was run from a Docker container using the following options: *-FusionInspectorvalidate, -examine_coding_effect,* and *–denovo_reconstruct.* Due to the low coverage of scRNA-seq samples, both filtered fusion detection results

and preliminary results were combined and processed, in which we only filtered for potential *TMPRSS2-ERG* fusion events.

*Bulk RNA-sequencing validation.* Two publicly available bulk RNA-sequencing PCa datasets were used to test the correlation between the PCa-enriched cell-state signatures and *AR* signaling, including Prostate Adenocarcinoma (TCGA[25], Firehose Legacy) dataset (*N* = 499, available at https://www.cbioportal.org/study/summary?id=prad_tcga), Metastatic Prostate Cancer, SU2C/PCF Dream Team (SU2C[49], PNAS 2019) dataset (*N* = 266, available at https://www.cbioportal.org/study/summary?id=prad_su2c_2019). For each dataset, mRNA expression was downloaded and normalized. Signature scores of *AR* signaling (Hallmark androgen response pathway), BE, LE, and club cell states as well as *ERG*+ and *ERG*- tumor cell signature scores were computed for each sample via ssGSEA analysis. Samples in each dataset were rank-ordered by the *AR* signature scores and heatmaps were generated using the customized scripts. To test the correlation between *AR* signature scores and each cell-state signature score, we computed the information coefficient and corresponding *P* values followed by FDR correction to evaluate the correlation. For tumor cell signatures, we computed the correlations between the *ERG* fusion status from each dataset and the signature scores of *ERG*+ and *ERG*- tumor cell gene sets we had previously generated. We ranked ordered the bulk RNA-seq samples according to whether or not the *TMPRSS2-ERG* fusion was detected and plotted the *ERG*+ and *ERG*- tumor cell signature score heatmaps. Information coefficient (IC), *P* values, and FDR *q* values were computed.

*Immune cell analysis.* T-cell and myeloid cell populations were sub-clustered separately following a similar pipeline as described above. For T-cells, 23 PCs and a resolution of 1.5 were selected for the clustering. For myeloid cells, 27 PCs and a resolution of 1.5 were selected. Cell clusters were annotated by a dot plot showing the top ten most expressed genes in each cluster.

Monocytes, macrophages, neutrophils, and eosinophils were identified and annotated based on the automated SingleR analysis[19]. M1, M2 macrophage phenotypes, tumor-associated macrophages, and two types of myeloid-derived suppressor cells were identified using documented markers from previous studies.

**Reporting summary**. Further information on research design is available in the Nature Research Reporting Summary linked to this article.

## Data availability

Raw single-cell RNA-sequencing FASTQ files and gene expression matrices files generated in this study have been deposited in the Gene Expression Omnibus (GEO) under accession number GSE176031. Whole-exome sequencing FASTQ files of the three primary prostate cancer patients in this study have been deposited in the European Genome-phenome Archive (EGA) under accession number EGAS00001005685. The Henry et al. normal human prostate scRNA-seq dataset[9] is available in the NCBI GEO database under accession number GSE117403. Two previously published datasets were processed and used for testing our PCa-associated club cell characterization. Raw gene count matrices files were downloaded from GSE146811 [Kauthaus et al. dataset[45]] and GSE141445 [Chen et al. dataset[46]]. RNA-seq data for signature validation are available at cBioportal [TCGA, SU2C]. Curated gene-set collections in this study (Hallmark, C2CP, and C2CGP) can be found at MsigDB. Source data are provided with this paper.

## Materials availability

This study did not generate new unique reagents.

## Code availability

All software algorithms used for analysis are available for download from public repositories. All code used to generate figures in the manuscript will be made available in the following Github repository: https://github.com/angelussong/scRNA-seq-Analysis-of-Prostate-Cancer-Samples. Source data are provided to generate figures in the manuscript.

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

## Acknowledgements

This work was in part supported by Searle Scholars Program (A.K.S.), Beckman Young Investigator Program (A.K.S.), Sloan Fellowship in Chemistry (A.K.S.), and the Pew-Stewart Scholars Program for Cancer Research (A.K.S.), Department of Defense grant W81XWH-17-PCRP-HD (F.W.H.), National Institutes of Health/National Cancer Institute grant P20 CA233255 (F.W.H.), Prostate Cancer Foundation Challenge Award 19CHAS03 (F.W.H.) and Young Investigator Award (F.W.H.), Benioff Initiative for Prostate Cancer Research (F.W.H.). We thank Norma Neff and the Chan Zuckerberg Biohub for help with sequencing.

## Author contributions

Conceptualization: H.S., M.R.C., A.K.S., and F.W.H; methodology: H.S., H.N.W., P.A., E.A.C., M.H.W., B.W., and F.W.H.; formal analysis: H.S., H.N.W., P.A., M.H.W., J.X., A.K.S., and F.W.H.; investigation: H.S., H.N.W., P.A., M.H.W., J.X., H.Y., K.L.L., B.A.S., and F.W.H.; data curation: H.S. and J.X.; writing—original draft: H.S. and F.W.H.; writing—review & editing: H.S., H.N.W., P.A., M.H.W., J.X., F.Y.F., P.R.C, M.R.C., A.K.S., and F.W.H.; resources: H.Y., P.C., B.W., M.R.C., A.K.S., and F.W.H.; supervision: F.W.H.

## Competing interests

A.K.S. reports compensation for consulting and/or SAB membership from Merck, Honeycomb Biotechnologies, Cellarity, Hovione, Repertoire Immune Medicines, Ochre Bio, Third Rock Ventures, Relation Therapeutics, and Dahlia Biosciences. F.Y.F. reports compensation for consulting and/or SAB membership from Astellas, Bayer, Blue Earth Diagnostics, Celgene, Genentech, Janssen Oncology, Myovant, Roivant, Sanofi, PFS Genomics, and SerImmune. The remaining authors declare no competing interests.
