## [Peer Review File · Nature Communications]

REVIEWER COMMENTS

Reviewer #1 (Remarks to the Author): Expert in prostate cancer genomics

General comments: The manuscript from Song et al. is the first report on single cell analysis of local prostate cancer. We believe that the findings of this work will serve as an important first step towards the description of the cellular atlas of localized prostate cancer. The experiments and subsequent analyses and discussions are executed according to the state of the art.

Prostate cancer research is running behind on the single cell analysis and work like described here will enhance and inspire many current and future work but also in the long run be translated to clinical practice. While the high level of heterogeneity of PCa is again illustrated by this work, one can also understand that this is the level that personalized medicine can and will be developed. The number of samples is limited, but it remains a first in class paper.

Specific comments and questions:

1. Supplementary tables are not easy to read and interpret, please include in the supplementary file.
2. It seems that the number of cells that were analyzed from biopsy samples is quite low. Please indicate the absolute cell counts per sample. Also, some analysis are based on very low amounts of cells and the number of tumor samples is also low. This is a clear weakness and could be addressed more.
3. Therapeutic implications could be discussed better: immune checkpoint inhibitors in what setting, what trials would be interesting to run?
4. Line 114: no difference in cell composition between T and non T samples. This should be defined better: what celltypes did you analyze to come to this statement.
5. Figure 1d: it seems like there are differences between the matched RP specimens and others on cellular composition. Please discuss.
6. Line 132-133: Would it be possible to find basal subtypes in your analysis? Using the Wallace Prostate cancer signatures, I would estimate some cancer cells have basal signatures.
7. Line 410: Tumor microenvironment is more than T cells alone, did you investigate the other cells making up the TME? If so where there any changes between Cancer/non Cancer? S3F seemingly shows some pronounced differences, is it possible/relevant to go more into detail? Are these population/batch effects or actual differences?
8. Line 442: It would be nice to see a comparison on specific cell types/ states (in more detail)
9. Line 850: 21743 cells seems rather low for sequencing 18 samples + organoid cultures. Please comment and provide cell counts per sample.

Reviewer #2 (Remarks to the Author): Expert in prostate cancer genomics and single-cell RNA-seq

Song et al perform a single cell study of human prostate cancer, characterizing cellular heterogeneity and using bioinformatics to suggest that club cells are progenitors for prostate luminal epithelia. ERG- and ERG+ tumors are shown to exhibit a highly similar tumor microenvironmental response while organoids appear to diverge in cellular content from the parental tumor. The manuscript is clearly written and informed and attempts to address the highly contentious notion of a prostate tumor stem cell.

This is an incredibly important data set; however, the underlying assumptions regarding the analysis of these data are problematic. The major issue is the averaging of gene expression across a cluster in normal vs. diseased when it could be that luminal epithelia in cancer may lose differentiation to a point that they begin to cluster with club or basal epithelia rather than basal or club cells turning on prostate luminal cell differentiation. CNV analysis should show whether basal or club cells with some luminal markers contain copy number variations consistent with luminal tumor epithelia.

1. Hillock and club cells are not prostate, they are prostatic urethra. This is an important distinction as

they are not Nkx3.1+.

2. There is no direct evidence in humans as of yet that hillock or club cells are progenitors for prostate luminal cells. Actually, the opposite evidence is shown in mice where androgen-independent Sca-1+/Psc+ urethral luminal epithelia (analogous to club/hillock urethral luminal epithelia in human) are a separate, self-sustaining lineage from the Nkx3.1+/Pbsn+ prostate luminal cell. (PMID 32497356 and 32627901) Also, clonal analysis shows that Sca-1+/Psc+ urethral luminal epithelia in the proximal prostate do not regenerate Nkx3.1+ prostate luminal epithelia in the mouse, rather that prostate luminal epithelia acquire a survival signature that resembles Sca-1+/Psc+ cells (PMID 32355025). Based on these data, the clustering of NKX3.1+/KLK3+ epithelia with club cells is likely a survival adaptation – i.e., they are likely de-differentiated prostate luminal epithelia that take on enough similar gene expression to look like (and cluster with) club cells. Based on the gene expression of cluster 0 in Figure 3 (PCa club cells), it appears that these are just prostate luminal cells with low androgen signaling that clustered with club cells (similar to the adaptation of PLCs in castrated mice (PMID 32355025)). This is the problem with identifying the epithelial lineage with SingleR (suppl Table 1) and then using a cluster-based analysis in the second tier of analysis to identify epithelial subtypes. Here you will see that dimensionality reduction may not be perfectly placing every cell of a certain type into 1 cluster. You need to do analysis of club genes in cluster 0, not just prostate luminal genes in Fig 3. Additionally, club cell genes (SCGB3A1, etc) need to be shown side-by-side with luminal cell genes in Supp fig 2. Basically, you need to make sure that the decrease in SCGB3A1 in the club cluster in prostate cancer is not simply due to the co-clustering of prostate luminal cells with low androgen signaling that look more like club cells than normal luminal cells. Would be important to perform SingleR in single cell mode after lineage identification of epithelia rather than relying on club cell identification in cluster mode.

3. The same issue is evident in Figure 4 with Basal epithelia. Need to show basal and luminal markers side by side in each cluster to make sure de-differentiated tumor luminal epithelia are not clustering with basal rather than basal epi in the tumor turning on normal luminal markers. You would assume that they should share markers of both lineages.

4. Given that patients treated with ADT (PMID 32355025) or 5ARI (PMID 32497356) show an increase in club cells, it is important to make clear whether the patients were treated with ADT (I saw 1 was treated with a 5ARI). If they are not treated with ADT, please state so clearly.

5. If club cells are a progenitor for prostate tumor cells, then where is the CNV analysis that shows that they have mutations? This is essential to show. If they do not have evidence of mutations, then they are likely just another cell in the microenvironment and not a prostate tumor cell progenitor. I would also be careful in Supp Fig 2 about calling genes that are normally expressed in club cells (PSCA, PIGR) 'progenitor' markers...

6. Another example of a potentially faulty assumption is the pseudotime analysis. Using a 'stem cell' signature as the starting point assumes that club cells are progenitors.

7. The spheroid culture analysis is very interesting, but you need to be careful about making the statement that they accurately recapitulate tumor cell heterogeneity states. Prostate tumor cells are well known to have difficulty growing ex vivo. The club and hillock and basal epithelia may just survive better in spheroid culture.

If it is true that basal and club cells do not have any copy number variations, the thrust of the paper might not be about their potential as tumor progenitors but an analysis of gene expression changes dictating their role in supporting the tumor might become more prominent.

Reviewer #3 (Remarks to the Author): Expert in prostate cancer organoids

I have been asked to specifically review the organoids section (Figure 8) of this study. It appears that all analyses and conclusions are expertly performed. I have a single major point of concern. It is a general experience that -unless growth conditions are adapted to only allow malignant cells to expand, contaminating wt epithelial cells rapidly overgrow the cancer cells. For colon cancer, one can simply remove Wnt/Rspondin from the medium, selecting for APC -and other Wnt pathway- mutants. This will

lead to the almost instantaneous death of non-cancer epithelial cells. For pancreas carcinomas, EGF can be left out (selecting for Ras mutants) or Nutlin-3 can be added (selecting for P53 mutants).

For prostate cancer, this phenomenon has posed a major problem. Given the way that the tumors grow rather diffusely in the prostate, it is virtually impossible to start from a 'pure' cancer sample. No selection method for prostate cancer organoids has been reported to date. Within 2-3 passages, contaminating wt cells will almost invariably dominate the culture. It is imperative to confirm that the growing organoids are indeed cancer cells. This can be done by molecular analysis of the oncogene alterations that are present in the primary tumor, as these will have to be retained in the organoids. Another approach could be to look at the retention of copy number alterations (which will also give an indication of the ratio cancer vs wt cells). RNA- or surface marker analysis unfortunately do not resolve this issue.

Because of the expression analyses that are presented in Figure 8, the mere presence of oncogene alterations/fusions will not be good enough: the overwhelming majority of the cells will have to be derived from the malignant clone in order to interpret these expression data as coming from the cancer cells.

Reviewer #4 (Remarks to the Author): Expert in prostate cancer and prostate cancer organoids

The manuscript entitled "Single-cell analysis of human primary prostate cancer reveals the heterogeneity of tumor-associated epithelial cell states" by Song et al. focuses on the understanding of tumor microenvironment and cellular states associated with prostate cancer. Overall, the concept of generation single cell RNA sequencing data to understand the complexity of cancer is sound. The authors have collected 6 prostate cancer biopsies from 3 patients, 4 radical prostatectomies with tumor-only and 4 radical prostatectomies with matched normal. For a total of 11 tumor samples and 4 normal. How do the authors know these contain tumor material? From figure 1a it is not clear how big the specimen from the RP is, this is relevant given that tumor content can vary. Do the authors have the histopathological assessment of the tissue they have sequenced or derived organoids from? In their assessment of the luminal phenotype the authors use markers such as KLK3, ACPP and MSMB, surprisingly NKX3.1 is not included. What is the reason for this? This is the first marker of luminal cells of the prostate. The authors make a point to highlight that the different sampling methodologies yield equivalent cell type composition and test whether biopsies from two anatomical regions identify similar cell types. Which anatomical regions and what is the purpose of this?

The paragraph "epithelia cell cluster reveal...." is very confusing. The authors start with the description of the identified clusters and the report about the roles of PIGR, MMP7 and CP identified by other. The authors identify 4 ERG positive clusters, one only ERG positive and 3 double positive (ETV1,4 or 5) and 7 ERG negative but positive for SPON2....and? How do they define that ERG negative tumors contain tumor specimen?

Relative to figure 2d in which the authors compute the top 10 biomarkers for each cell types, the LE do not express or not include NKX3.1? Interestingly, the LE subpopulation seems to be the most responsive to androgens, but the BE is predicted the stem cell, being enriched for the stem cell signature. How do the authors discuss this based on the evidence that stem cells and cancer stem cells can be both of luminal and basal origin and both sensitive to androgen levels?

The authors point out that they cannot identify co-expression of luminal markers that have been defined to characterize a luminal progenitor cell type, concluding that the presence of two of the markers in their club cell subtype is sufficient to identify this population with the luminal progenitor stem cells. In order to support this strong statement, the authors should use the derived organoid and perform molecular in vitro and in vivo characterizations.

The authors hypothesize that the identified club cells play a role in carcinogenesis. They go through extensive bioinformatic analysis to support their hypothesis. As a comment GSEA is not a “functional” validation, it may be an in silico validation. They additionally generate organoids and characterize their cell content however there are no functional experiments done that will support their hypothesis.

As a comment of the aspect of organoid formation, given the huge debate in the literature regarding the generation of cancer organoids from primary prostate cancer, the burden lays on the authors to prove that the tissue and organoids do indeed contain tumor tissue. Therefore the CNV calling should be validated orthogonally against scDNA-seq, WES or Selected gene panel. Moreover, a much more extensive characterization of the organoids is required, including histopathological assessment, marker expression and localization. In particular of the markers that the authors pick to determine their subpopulations. Finally, one is least wondering why the authors go through the trouble to generate organoids if they do not use them to their full potential for, to the least, in vitro validation of their hypothesis.

The execution of the computational pipelines looks very standard, usage of Seurat, STAR Fusion and CNVInfer. My concern is, scRNA-seq suffers from shallow sequencing and the authors have used CNVInfer tool to look for CNV. The point is, small focal changes will not be captured as compared to WGS or exome. In addition, if one feeds in a list of expected copy number alterations from bulk RNA-seq or WGS a better resolution/capture of CNV can be achieved (I did not see this part being mentioned).

Reviewer #5 (Remarks to the Author): Expert in immunogenomics and single-cell RNA-seq

This work contains satisfactory cohort and scRNA-seq data. The data generated for this analysis is sufficient to clearly determine that the ERG status has significant effect the immune and stromal compartments. However, few comments need to be addressed prior to publication:

Moderate:

The authors suggest that the stratification of the T-cell populations based on ERG status showed two CD4 T-cell clusters that were differentially enriched. While this is correct there are additional differences which were overlooked for example CD8 cluster 1 and CD8 cluster 2 shows different internal distribution. Thus, to avoid biases due to clustering granularity it would be recommended to perform additional re-clustering of these populations. Such analysis could directly affect the authors' conclusion regarding the immune microenvironment in ERG+/-

In addition, the authors claim that CD4 and CD8 T-cells associated with ERG- tumor cells represented a more exhausted and cytotoxic phenotype, yet they show only a couple of markers for these T cell states. Are these exhausted or activated cells? An additional signature enrichment analysis will further strengthen their claim. Here also please note that - CCR7+, CD69: higher in ERG+, granzyme A and B: higher in ERG- and MKI67 is higher in ERG-.

Finally, the analysis in 7e of the common pathways upregulated in cancer, CD4 cells and stromal cells is not clear. In the T cells, not only CD4 cells show a difference, in the stromal cells they emphasized the fibroblast and the cancer cells have multiple clusters.

Minor:

1. Fig 1a – Please provide the marker gene list used for the annotation (for specific T-cells)

subsets:CD8+, CD4+ Treg).

2. Fig 1a – Please keep the same colors scheme ERG- (blue) and ERG+ (red) in a,b,c,d. (it is currently different between the UMAP and bar-graphs).

3. Line 419, I could not find the data for upregulating the genes in CD4+ cells (FOS,JUN, CXCR6 etc.)

4. Line 434 need to add fig 7d.

5. Line 437, It is not clear why the authors analyzed all stomal cells for the analysis, rather than just the fibroblast, which they have shown to have different distribution between ERG+ and ERG- (similar to CD4+ cells).

6. In fig 7a as in fig 7c, please position the legends at the same place (for example in 7c it could confuse the readers to think that ERG- is in smooth muscles and ERG+ in fibroblast)

7. Line 441: to change from fig 7f to e.

8. Line 444: to change from 7g to f.

9. Line 448: to change from 7h to g.

10. Supplemental fig 6 – Please change the mode of visualization, or add mean, so we can visualize up or downregulated genes, in the current violin plot version, we cannot appreciate changes in genes that are not described in the text. It would be interesting to know if the markers of the same family without significant differences are increased or decreased in the same way (ex: HAVCR2, LAG3, Tigit in CD4+)

11. Supplementary fig 7d and e: be consistent with the colors of the legends (M0, M1, M2 macrophages)

Reviewer #1 (Remarks to the Author)

General comments: The manuscript from Song et al. is the first report on single cell analysis of local prostate cancer. We believe that the findings of this work will serve as an important first step towards the description of the cellular atlas of localized prostate cancer. The experiments and subsequent analyses and discussions are executed according to the state of the art. Prostate cancer research is running behind on the single cell analysis and work like described here will enhance and inspire many current and future work but also in the long run be translated to clinical practice. While the high level of heterogeneity of PCa is again illustrated by this work, one can also understand that this is the level that personalized medicine can and will be developed. The number of samples is limited, but it remains a first in class paper.

We sincerely appreciate the positive feedback from the Reviewer.

Specific comments and questions:

1. Supplementary tables are not easy to read and interpret, please include in the supplementary file.

As the Reviewer suggested, we have now reformatted some of our supplementary tables and the improved version is now included in the supplementary files.

2. It seems that the number of cells that were analyzed from biopsy samples is quite low. Please indicate the absolute cell counts per sample. Also, some analysis are based on very low amounts of cells and the number of tumor samples is also low. This is a clear weakness and could be addressed more.

We thank the Reviewer for this comment. We now have provided the raw cell counts for each sample before and after the preprocessing in the supplementary tables (Supplementary table 1). We agree that some of our conclusions were based on relatively small cell counts but note that two main populations of our focus, club cells and T-cells, were comprised by more than one patient and that 9 of 11 patients had at least 30 T-cells and 30 club cells. Even though the CD4 T-cell population represented 18.5% of all the T-cells, these CD4 T-cells were detected in all patients. Therefore our analysis of the difference between *ERG+* and *ERG-* CD4 T-cells was not confounded by any single patient.

3. Therapeutic implications could be discussed better: immune checkpoint inhibitors in what setting, what trials would be interesting to run?

We thank the Reviewer for this comment and agree this is an interesting point to discuss. We have now added to the discussion the following statement: “While immune checkpoint inhibition (ICI) has had limited efficacy in advanced castration-resistant prostate cancers, we speculate that there may be targeted trials in *ERG*-negative hormone-sensitive prostate cancers that may be more beneficial with ICI or other emerging immunotherapies that are currently in clinical trials.”

4. Line 114: no difference in cell composition between T and non T samples. This should be defined better: what celltypes did you analyze to come to this statement.

We thank the Reviewer for pointing this out. To elaborate, we analyzed the non-tumor epithelial cells (BE, LE and club cells), stromal cells (endothelial, fibroblast and smooth muscle), and immune cells (T-cells and myeloid cells) and found no significant differences in their composition. We have revised the statement accordingly in the results section: “We also compared the cell type composition of epithelial cells, stromal cells (endothelial, fibroblasts and smooth muscle), and immune cells (T-cells and myeloid cells)”.

5. Figure 1d: it seems like there are differences between the matched RP specimens and others on cellular composition. Please discuss.

Even though the proportions of epithelial cells in the matched RP specimens were higher than other samples, the non-parametric statistical test yielded p-values of greater than 0.05 therefore the differences were not significant.

6. Line 132-133: Would it be possible to find basal subtypes in your analysis? Using the Wallace Prostate cancer signatures, I would estimate some cancer cells have basal signatures.

We thank the Reviewer for this question about the possibility that there might be tumor cells with basal signatures. In the analysis, we did use the Wallace Prostate cancer signature for tumor cell identification but we did not detect a strong correlation between this signature and a basal signature in the tumor cells as we tested the basal markers *KRT5* and *KRT15* expression in the tumor cells. We found 2.9% of tumor cells were *KRT5+* and 4.2% were *KRT15+*, representing only a rare sub-population. Then we re-clustered only the tumor cells (*ERG+* and *ERG-* tumor cells) into 14 clusters (see below). Using a basal cell signature previously developed from bulk-RNA seq samples (Smith et al., PNAS, 2015, doi: <https://doi.org/10.1073/pnas.1518007112>), only 31 cells showed a high basal signature score of >0.2. Therefore, we did not observe any cluster of tumor cells with an enrichment of the basal cell signature. In the revised manuscript, we have added a sentence in the results section reporting this: “No tumor clusters detected within our dataset showed an enrichment of a BE signature.”

7.Line 410: Tumor microenvironment is more than T-cells alone, did you investigate the other cells making up the TME? If so where there any changes between Cancer/non Cancer? S3F seemingly shows some pronounced differences, is it possible/relevant to go more into detail? Are these population/batch effects or actual differences?

Besides T-cells, we also investigated myeloid cells and stromal cells between the paired tumor samples and paired normal samples. However, due to the high tumor content in these patients, the total number of myeloid and stromal cells in the paired tumor samples and paired normal samples were relatively low. Therefore, we were not able to make any generalizable conclusions.

When studying the epithelial cells in our dataset, we integrated the epithelial cells with normal healthy prostate epithelial cells from the Henry et al. study. As we captured cells from both normal and PCa samples in all sub-clusters this suggests that these differences are unlikely to be batch effects.

8.Line 442: It would be nice to see a comparison on specific cell types/ states (in more detail)

We thank the Reviewer for this suggestion. In the manuscript, since we aimed at identifying all coordinated pathway changes in the epithelial cells between *ERG+* and *ERG-* patients to identify transcriptomic differences that tumor cells might give rise to differential responses in the tumor microenvironment, we only reported that “GSEA analysis between *ERG+* and *ERG-* patients for tumor cells, BE, non-malignant LE and club cells using the C2CP gene set collection did not detect any common pathway changes shared by these epithelial cell types. BE cells in *ERG-* patients were found to be significantly upregulated ($q < 0.05$) in 19 pathways in all 2500 C2CP collection, but no pathway was found to be significantly downregulated ($q < 0.05$). For club cells, no pathway was found to be significantly upregulated or downregulated in *ERG-* patients ($q < 0.05$)”. We have now included the C2CP collection GSEA report for BE and club cells between *ERG+* and *ERG-* patients in a supplemental table (Supp Table 5).

Furthermore, in addition to Hallmark GSEA analysis we reported in the manuscript, we ran GSEA analysis for C2CP collection between club cell state 0 and other club cells, as well as BE cell state 6 and other BE cells in our dataset. The table results for these GSEA analysis are now provided in the updated Supplemental table 5.

9.Line 850: 21743 cells seems rather low for sequencing 18 samples + organoid cultures.
Please comment and provide cell counts per sample.

In the manuscript, we analyzed a total number of 36,816 cells (21,743 cells from 18 tumor tissue samples and 15,073 cells from organoid samples). We have added the information of cell counts per sample in the revised supplemental tables (Supplementary table 1).

Reviewer #2 (Remarks to the Author)

Song et al perform a single cell study of human prostate cancer, characterizing cellular heterogeneity and using bioinformatics to suggest that club cells are progenitors for prostate luminal epithelia. ERG- and ERG+ tumors are shown to exhibit a highly similar tumor microenvironmental response while organoids appear to diverge in cellular content from the parental tumor. The manuscript is clearly written and informed and attempts to address the highly contentious notion of a prostate tumor stem cell.

This is an incredibly important data set; however, the underlying assumptions regarding the analysis of these data are problematic. The major issue is the averaging of gene expression across a cluster in normal vs. diseased when it could be that luminal epithelia in cancer may lose differentiation to a point that they begin to cluster with club or basal epithelia rather than basal or club cells turning on prostate luminal cell differentiation. CNV analysis should show whether basal or club cells with some luminal markers contain copy number variations consistent with luminal tumor epithelia.

1. Hillock and club cells are not prostate, they are prostatic urethra. This is an important distinction as they are not *Nkx3.1+*.

We thank the Reviewer for this comment. In a previous single-cell study of normal healthy human prostates (Henry et al. 2018), the authors observed separate clusters of hillock and club cells in the urethra zone using different gating methods. Using the gene markers of their club cell analysis, we were also able to detect such cells in the tissues in our analysis. In both our dataset and the normal dataset, *NKX3-1* was detected predominantly in luminal cells therefore we considered it as a luminal marker. To clarify this point, we have revised figure 2 and now added *NKX3-1* as one of the luminal markers and observed that compared to luminal cells and tumor cells, club cells express *NKX3-1* at a much lower level.

2. There is no direct evidence in humans as of yet that hillock or club cells are progenitors for prostate luminal cells. Actually, the opposite evidence is shown in mice where androgen-independent *Sca-1+/Psc+* urethral luminal epithelia (analogous to club/hillock urethral luminal epithelia in human) are a separate, self-sustaining lineage from the *Nkx3.1+/Pbsn+* prostate luminal cell. (PMID 32497356 and 32627901) Also, clonal analysis shows that *Sca-1+/Psc+* urethral luminal epithelia in the proximal prostate do not regenerate *Nkx3.1+* prostate luminal epithelia in the mouse, rather that prostate luminal epithelia acquire a survival signature that resembles *Sca-1+/Psc+* cells (PMID 32355025). Based on these data, the clustering of *NKX3.1+/KLK3+* epithelia with club cells is likely a survival adaptation – i.e., they are likely de-differentiated prostate luminal epithelia that take on enough similar gene expression to look like (and cluster with) club cells. Based on the gene expression of cluster 0 in Figure 3 (PCa club cells), it appears that these are just prostate luminal cells with low androgen signaling that clustered with club cells (similar to the adaptation of PLCs in castrated mice (PMID 32355025)). This is the problem with identifying the epithelial lineage with SingleR (suppl Table 1) and then using a cluster-based analysis in the second tier of analysis to identify epithelial subtypes. Here you will see that dimensionality reduction may not be perfectly placing every cell of a certain type into 1 cluster. You need to do analysis of club genes in cluster 0, not just prostate luminal genes in Fig 3. Additionally, club cell genes (*SCGB3A1*, etc) need to be shown side-by-side with luminal cell genes in Supp fig 2. Basically, you need to make sure that the decrease in *SCGB3A1* in the club cluster in prostate cancer is not simply due to the co-clustering of prostate luminal cells with low androgen signaling that look more like club cells than

normal luminal cells. Would be important to perform SingleR in single cell mode after lineage identification of epithelia rather than relying on club cell identification in cluster mode.

We thank the Reviewer for these insights and for proposing another possibility of the identities of these club cells we describe in our analysis. As suggested, to further test if the cluster 0 of club cells were luminal cells with low androgen signaling, we revised the stacked violin plots in Figure 3 (see below, left panel) to show luminal markers and club cell markers side-by-side for our LE populations, club cluster 0 and other club cells. We also generated a large-scale heatmap (Supplemental Figure 5d) showing club markers and luminal markers side-by-side for all epithelial cells by their annotation (see below, right panel). Since SCGB3A1 is lowly expressed in club cells in our dataset, we showed SCGB1A1 instead which is also a club cell marker to show that SCGB1A1 in club cells are significantly higher than in luminal cells.

From these comparisons, we see that the club cells we annotated expressed significantly higher markers consistent with the club cells Henry et al. detected in urethra, and that while a small percentage of club cells express luminal markers, the expression level and percentage of expression are much lower than luminal cells. Therefore, we annotated these cells as club cells instead of luminal cells with lower androgen signaling, though with the single cell data in this study we cannot rule out the possibility that these are luminal cells that have lost luminal markers and transitioned to a club cell expression profile.

To further investigate the other possibility that these club cells might be luminal cells turning on club cell markers, we ran inferCNV analysis (see below, left panel) on all cells from our dataset using all the non-epithelial cells as reference and our annotated epithelial cells as observations. If these club cells are dedifferentiated luminal cells, on a global CNV level, we would expect to detect similar CNV profiles between these club cells and the non-malignant luminal population. However, when we imported the inferCNV score and computed the overall distributions for all epithelial populations, we saw that compared to luminal cells, the estimated CNV profile of club cells was not significantly different from basal cells (right panel), for example the amplification on chr4 and chr9, both of which were absent for non-malignant luminal cells and tumor cells, suggesting that these club cells were not, by lineage, a dedifferentiated luminal population.

We appreciate the good suggestion about singleR and ran singleR in single-cell mode for epithelial cells and club cells separately (see below), and due to the likely lack of detailed reference in the singleR library, all club cells were categorized as epithelial cells by singleR. In the revised manuscript method section, we added the following statement to highlight why singleR wasn't more informative in epithelial cell annotation: "However, when running singleR in single-cell mode for epithelial cells, and due to the lack of detailed reference in the singleR library, singleR could not identify detailed epithelial cell types. Therefore, manual annotation was required for epithelial cells."

Furthermore, to investigate club cells in prostate single cell data, we analyzed two publicly available scRNA-seq datasets of prostate cancer specimens (N = 8 patients, Karthaus et al., Science, 2020; N = 13 patients, Chen et al., Nat. Cell Biol. 2021). Using the club cell signature gene set we generated from our data, we identified distinctive club cell clusters in both datasets that are expressing significantly higher levels of *PIGR*, *MMP7*, *LTF* and *CP* compared to other epithelial cell types. We also found evidence of the PCa-enriched club cell state that is *NKX3-1+* in these two datasets. We have included this additional detailed analysis in the revised manuscript (Supplemental Figure 5).

3. The same issue is evident in Figure 4 with Basal epithelia. Need to show basal and luminal markers side by side in each cluster to make sure de-differentiated tumor luminal epithelia are not clustering with basal rather than basal epi in the tumor turning on normal luminal markers. You would assume that they should share markers of both lineages.

We agree with the Reviewer that some basal cells share markers of both basal and luminal markers. According to the unsupervised inferCNV analysis we show above, we assess that it is less likely that the basal cells (including basal cluster 6) in our dataset were de-differentiated luminal cells but acknowledge this is a possibility. Within basal cells, when we checked the expression of both luminal and basal markers in BE cluster 6 and other basal cells (see below), the expression of basal markers such as *KRT5*, *KRT15*, and *TP63* did not show any significant differences (Figure 4g). The only significant differences in BE cluster 6 was that it showed much higher luminal marker expression, suggesting that this BE cluster 6 was more likely to be a cluster of basal cells with expression of luminal markers. Therefore we revised the result section in the manuscript to clarify this: “Therefore, we compared the expression levels of BE and LE markers between BE cluster 6 and other BE cells, and found that the expression of basal markers such as *KRT5*, *KRT15*, and *TP63* did not show any significant differences and the only significant differences in BE cluster 6 was the higher expression of luminal markers”.

4. Given that patients treated with ADT (PMID 32355025) or 5ARI (PMID 32497356) show an increase in club cells, it is important to make clear whether the patients were treated with ADT (I saw 1 was treated with a 5ARI). If they are not treated with ADT, please state so clearly.

We thank the Reviewer for making this point and in the revised manuscript, we now state that of PCa patients in our analysis only one patient was treated with 5ARI (Methods, “Of these 11 patients, only one patient (patient 4) was treated with finasteride”). We found no significant enrichment of club cells in this patient compared to others.

5. If club cells are a progenitor for prostate tumor cells, then where is the CNV analysis that shows that they have mutations? This is essential to show. If they do not have evidence of mutations, then they are likely just another cell in the microenvironment and not a prostate tumor cell progenitor. I would also be careful in Supp Fig 2 about calling genes that are normally expressed in club cells (*PSCA*, *PIGR*) ‘progenitor’ markers...

We thank the Reviewer for the suggestion of conducting CNV analysis for club cells. We ran inferCNV analysis on all cells from our dataset using all the non-epithelial cells as reference and our annotated epithelial cells as observations. Besides the qualitative heatmap, we also imported the inferCNV score and computed the overall distributions for all epithelial populations. From the results, we saw that compared to luminal cells, the estimated CNV profile of club cells were more similar to basal cells, showing no significant differences. However, since the inferCNV analysis, by its nature, was an estimation of CNV events based on the transcriptomic differences, we could not definitively illustrate the mutation profiles of club cells.

We agree that these club cells might not be prostate tumor cell progenitors and could serve as another cell in the tumor microenvironment. Given the current read depth of single-cell RNA-seq samples, our current data cannot prove the progenitor cell hypothesis. Therefore, we have revised our manuscript to indicate the possibility that club cells may function as another cell in the tumor microenvironment. We also appreciate the Reviewer's comment about clarifying genes as 'progenitor' markers. In the revised manuscript, we have now modified the description of *PSCA* and describe it as a marker enriched in PCa samples in previous studies (Results, "of which *PSCA* was detected to be enriched in PCa").

6. Another example of a potentially faulty assumption is the pseudotime analysis. Using a 'stem cell' signature as the starting point assumes that club cells are progenitors.

We apologize for the unclear description of the pseudotime analysis in our study. To validate our pseudotime analysis, we ran the trajectory analysis on the epithelial cells using the "slingshot" package which does not require specifying a starting point for the trajectory to avoid any pre-imposed assumption (See below). This result is consistent with our manuscript results and the Henry et al. study such that BE cells give rise to club cells, LE and both *ERG+* and *ERG-* tumor cells in our dataset. This provides support of our interpretation of the role of club cells in our dataset. We added the following text in the revised manuscript "We repeated the pseudotime analysis in an unsupervised manner without specifying the starting point and the pseudotime trajectory was consistent that BE cells give rise to club cells, LE and both *ERG+* and *ERG-* tumor cells in our dataset".

Furthermore, to perform the pseudotime analysis in a more unsupervised manner using ScanPy, we selected a different BE cell as the root cells for calculating the pseudotime trajectory as was done in the Henry et al. study, and repeated the test five times. In all tests, the BE population remained as the starting point for pseudotime, followed closely by the club population, suggesting that club cells arise later than BE cells in the differentiation timeline and that BE give rise to the other epithelial cell types. This also supports our interpretation of the roles of club cells such that they are more transcriptomically similar to BE compared to LE such that it is possible that club cells may harbor progenitor-like features.

To clarify our analysis and description in the paper, we now have revised the section of pseudotime analysis (Methods, "The diffusion pseudotime for each cell was calculated using SCANPY's `sc.tl.diffmap()` and `sc.tl.dpt()` with the root cluster as the BE cluster and then was plotted on the PAGA initialized embedding. We then visualized the gene marker changes along the pseudotime by cell type using `sc.pl.paga_path()`". Results, "Using BE cells as the starting point consistent with a previous scRNA-seq study, we plotted the diffusion pseudotime trajectory and observed that tumor cells and LE cells (KLK3+) were later than club cells (PIGR+, LTF+ and PSCA+) and KRT5+ BE cells (Supplemental Figure 5a,b), and that there was no significant difference in the computed pseudotime between BE and club cells (Supplemental Figure 5c)." and modified the supplementary figure 4 (see below) accordingly to reflect this improved analysis.

7. The spheroid culture analysis is very interesting, but you need to be careful about making the statement that they accurately recapitulate tumor cell heterogeneity states. Prostate tumor cells are well known to have difficulty growing *ex vivo*. The club and hillock and basal epithelia may just survive better in spheroid culture.

We agree with the Reviewer on this suggestion and acknowledge the possibility that the reason we captured fewer luminal cells could be attributed to the fact that club, hillock, and basal cells may survive better *in vitro*. In the revised manuscript, we rephrased the conclusion to describe our observations of epithelial cell types and cell states within organoid samples by stating "PCa-enriched club cell states were also observed in organoid samples" (Results), and "Furthermore, we showed that the cell states found in tumor tissues could also be identified within *in vitro* organoid cultures grown from tumor specimens" (Discussion) to avoid overstatement.

If it is true that basal and club cells do not have any copy number variations, the thrust of the paper might not be about their potential as tumor progenitors but an analysis of gene expression changes dictating their role in supporting the tumor might become more prominent.

We thank the Reviewer for pointing this out and we agree this is a plausible interpretation. We have now revised the interpretation of the role of club cells in our manuscript accordingly. The following changes are now made in the revised manuscript. Abstract, line 41-42: “We identify a population of tumor-associated club cells that may be associated with prostate carcinogenesis.”

Discussion, line 627-630: “Based on our gene signature analysis, our results suggest that PCa club cells are more androgen responsive overall and harbor a highly androgen-responsive cell state that may play a supporting role for the overall androgen responsive cellular milieu of prostate cancer.”

Reviewer #3 (Remarks to the Author): Expert in prostate cancer organoids

I have been asked to specifically review the organoids section (Figure 8) of this study. It appears that all analyses and conclusions are expertly performed. I have a single major point of concern. It is a general experience that -unless growth conditions are adapted to only allow malignant cells to expand, contaminating wt epithelial cells rapidly overgrow the cancer cells. For colon cancer, one can simply remove Wnt/Rspondin from the medium, selecting for APC - and other Wnt pathway- mutants. This will lead to the almost instantaneous death of non-cancer epithelial cells. For pancreas carcinomas, EGF can be left out (selecting for Ras mutants) or Nutlin-3 can be added (selecting for P53 mutants).

For prostate cancer, this phenomenon has posed a major problem. Given the way that the tumors grow rather diffusely in the prostate, it is virtually impossible to start from a 'pure' cancer sample. No selection method for prostate cancer organoids has been reported to date. Within 23 passages, contaminating wt cells will almost invariably dominate the culture. It is imperative to confirm that the growing organoids are indeed cancer cells. This can be done by molecular analysis of the oncogene alterations that are present in the primary tumor, as these will have to be retained in the organoids. Another approach could be to look at the retention of copy number alterations (which will also give an indication of the ratio cancer vs wt cells). RNA- or surface marker analysis unfortunately do not resolve this issue.

Because of the expression analyses that are presented in Figure 8, the mere presence of oncogene alterations/fusions will not be good enough: the overwhelming majority of the cells will have to be derived from the malignant clone in order to interpret these expression data as coming from the cancer cells.

We sincerely appreciate the Reviewer’s consideration of the analysis in our manuscript and fully agree on the comment that within early passages, wild-type cells will dominate the culture therefore the tumor cells we identified in the organoid analysis required oncogene alterations or copy number analysis to confirm their identity.

To address this issue, we ran a variant caller on the organoid scRNA-seq samples and filtered for potential somatic mutations. As a result, the organoid sample with the dominant contribution to the tumor population was detected with *FOXA1*, *KMT2C* and *MTOR* mutations despite the shallow sequencing of scRNA-seq. And when we ran the copy number estimation test via inferCNV on the organoid cells, we found that using the non-tumor cells as reference, tumor cells in the organoid showed a copy number amplification on chr11 and chr17 (see below), of which the amplification on chr11 was consistent with the profile of both *ERG+* and *ERG-* tumor cells in the copy number analysis we previously conducted on the tissue samples. Even though this copy number estimation was still based on the expression matrices of the samples, it still provides

support for our analysis results that tumor cells could be observed in this particular organoid sample.

Even though we understand that gene marker expression itself cannot definitively identify tumor cells, on a transcriptomic level we found that these cells harbored high-level expression of markers different from the other populations and this “tumor-like” population was distinctively consistent with the previously developed *ERG*+ tumor cell gene expression profile, showing upregulated *ERG*, *PCA3* and *AMACR* expression (see below).

In summary, we acknowledge that the annotation of these “tumor-like” cells requires further experimental validation, which is a limitation of our current study. These results are consistent with previous studies that primary prostate tumor cells are not easily cultured beyond a few passages. Our major aim for the organoid section was to identify basal cells and club cells in the organoids that showed the same markers as the tissue samples and we observed that cell-state transitions appear to occur in the organoid samples in vitro compared to the parent tissue samples. Therefore, in the discussion section of the revised manuscript, we added “We detected a population of “tumor-like”

cells largely from one patient organoid. It is possible that these “tumor-like” cells are present only initially in some organoids but dissipate after several passages.”

Reviewer #4 (Remarks to the Author): Expert in prostate cancer and prostate cancer organoids

The manuscript entitled “Single-cell analysis of human primary prostate cancer reveals the heterogeneity of tumor-associated epithelial cell states” by Song et al. focuses on the understanding of tumor microenvironment and cellular states associated with prostate cancer. Overall, the concept of generation single cell RNA sequencing data to understand the complexity of cancer is sound. The authors have collected 6 prostate cancer biopsies from 3 patients, 4 radical prostatectomies with tumor-only and 4 radical prostatectomies with matched normal. For a total of 11 tumor samples and 4 normal. **How do the authors know these contain tumor material?** From figure 1a it is not clear how big the specimen from the RP is, this is relevant given that tumor content can vary. Do the authors have the histopathological assessment of the tissue they have sequenced or derived organoids from?

We thank the Reviewer for raising this question as it is important to establish confidence that the cells we analyzed indeed contain tumor. For radical prostatectomy specimens, only prostates with palpable lesions were sampled. We have now provided a new Supplemental Figure 2 demonstrating that in 6 patients, there were visible lesions on preoperative MRI or ultrasounds corresponding to the palpable lesions that were sampled. These areas were re-examined by our pathology colleagues and were confirmed to contain cancer by histology. We have included representative images of the H&E stains that corresponds to each of the sampled lesions. For the remaining patients, the biobanking technician did not note specifically which lesion was sampled, therefore they were not included in this figure. Regarding the biopsy specimens, the area targeted by the biopsy was sampled multiple times. The pathologic assessment of the corresponding cores that were not processed for single cell analysis were examined pathologically and confirmed to contain cancer. This pathological assessment is now included in the revised manuscript (Supplemental Figure 2).

We further sought to histologically validate the presence or absence of *ERG* in tumor cells. We thus performed *ERG* immunohistochemistry in one *ERG*⁺ tumor and two *ERG*⁻ tumors and found that *ERG* staining in tumor cells is consistent with our single cell data (Supplemental Figure 2b). Nearly all tumor cells in the *ERG*⁺ tumor stain for *ERG* while none of the *ERG*⁻ tumors stain for *ERG*. There are scattered interstitial cells that do stain for *ERG* in the *ERG*⁻ tumors, but on closer examination many of these appear to be *ERG*⁺ endothelial cells, which is consistent with our single cell data. We added the following statement in the result section of the revised manuscript “*ERG* status was histologically validated in a subset of patients using immunohistochemistry (Supplemental Figure 2b)”.

In their assessment of the luminal phenotype the authors use markers such as *KLK3*, *ACPP* and *MSMB*, surprisingly *NKX3.1* is not included. What is the reason for this? This is the first marker of luminal cells of the prostate.

We thank the Reviewer for pointing out *NKX3-1* as a luminal marker. It was not among the top 10 differentially expressed genes (it was number 11) so we initially did not show it among the other luminal markers. We have now included *NKX3-1* in Figure 2 of the revised manuscript and the differentially expressed gene list in Supplemental Table 2.

The authors make a point to highlight that the different sampling methodologies yield equivalent cell type composition and test whether biopsies from two anatomical regions identify similar cell types. Which anatomical regions and what is the purpose of this?

The two biopsies from each patient were taken from different regions including left/right apex/mid/anterior regions. In the revised manuscript, we included this in the “Biopsy_composition” tab in Supplemental Table 1. Since these three patients were early-stage (T1c and T2A), urologists took biopsies from multiple regions to enrich for capture of tumor cells.

The paragraph “epithelial cell cluster reveal...” is very confusing. The authors start with the description of the identified clusters and the report about the roles of PIGR, MMP7 and CP identified by other. The authors identify 4 ERG positive clusters, one only ERG positive and 3 double positive (ETV1,4 or 5) and 7 ERG negative but positive for SPON2....and? How do they define that ERG negative tumors contain tumor specimen?

We apologize for the confusion in the description of the epithelial cell clusters. We have now revised this paragraph that describes characterizing epithelial subpopulations. Specifically, for the ERG+ tumor cells, we describe that “Therefore we tested cells for expression of ERG, ETV1, ETV4 or ETV5, and found that ERG expression was upregulated in four clusters which we annotated as ERG+ tumor cells (Figure 2b, Supplemental Figure 1a) but no cluster showed expression of ETV1, ETV4 or ETV5, suggesting that the six patients that contributed to these four ERG+ tumor cells harbored ERG fusion events.” (Results).

And to better characterize the ERG- tumor cells found in our dataset, we revised the manuscript text to state that “These 11 clusters included the four ERG+ tumor cell clusters we previously annotated and seven other clusters with no ERG expression therefore we annotated them as ERG- tumor cells (Figure 2c). The ERG- tumor cell clusters were characterized by the enrichment of at least one known PCa gene set signature as well as the higher expression of tumor markers such as SPON2 and PCA3 compared to non-tumor LE in our dataset (Figure 2b)” (Results).

Relative to figure 2d in which the authors compute the top 10 biomarkers for each cell types, the LE do not express or not include NKX3.1? Interestingly, the LE subpopulation seems to be the most responsive to androgens, but the BE is predicted the stem cell, being enriched for the stem cell signature. How do the authors discuss this based on the evidence that stem cells and cancer stem cells can be both of luminal and basal origin and both sensitive to androgen levels?

We thank the Reviewer for this question. In Figure 2d, we showed the heatmap of the top 10 biomarkers for each epithelial cell types. Those markers were selected by using the FindAllMarkers() in Seurat and rank ordered by their log2 fold change of expression compared to the other cell types. According to this calculation, NKX3-1 was not in the top 10 list even though it was still within the top 20 (number 11). Due to the limit of figure size, it was not shown in the heatmap (Figure 2d). We provided the list of differentially expressed genes for all our PCa epithelial cell types in Supp table 2.

We appreciate the Reviewer for pointing out that BE cells were more likely to be the stem cell population of prostate epithelia despite the fact that LE seemed to be more responsive to androgens. We performed pseudotime analysis in an unsupervised way using slingshot and found that BE was also the origin of the pseudotime trajectory and

that BE gave rise to LE as well as the two tumor cell populations in our dataset (below). Based on our analysis, we interpret that even though LE are more androgen responsive with higher expression of downstream AR targets, prostate cancer stem cells may be more likely to have a BE origin, and club cells, due to the similar position in the pseudotime trajectory as BE, might play a potential role in carcinogenesis. We added the following text in the revised manuscript “We repeated the pseudotime analysis in an unsupervised manner without specifying the starting point and the pseudotime trajectory was consistent that BE gives rise to club cells, LE and both *ERG+* and *ERG-* tumor cells in our dataset”.

The authors point out that they cannot identify co-expression of luminal markers that have been defined to characterize a luminal progenitor cell type, concluding that the presence of two of the markers in their club cell subtype is sufficient to identify this population with the luminal progenitor stem cells. In order to support this strong statement, the authors should use the derived organoid and perform molecular in vitro and in vivo characterizations.

The authors hypothesize that the identified club cells play a role in carcinogenesis. They go through extensive bioinformatic analysis to support their hypothesis. As a comment GSEA is not a “functional” validation, it may be an *in silico* validation. They additionally generate organoids and characterize their cell content however there are no functional experiments done that will support their hypothesis.

We thank the Reviewer for these points. The Reviewer makes a good point and we have revised the manuscript to modify our statements about potential progenitor role of club cells (please see our response to Reviewer 2 second comment) so that this is not overstated. In the revised manuscript, we edited the description of GSEA analysis into “To test the *in silico* functional role of this cell state” to avoid confusion. We appreciate the Reviewer’s suggestion and agree that functional experiments in the future will help us further test the role of club cells in carcinogenesis and we acknowledge are outside of the scope of the study here.

As a comment of the aspect of organoid formation, given the huge debate in the literature regarding the generation of cancer organoids from primary prostate cancer, the burden lays on the authors to prove that the tissue and organoids do indeed contain tumor tissue. Therefore the CVN calling should be validated orthogonally against scDNA-seq, WES or Selected gene panel. Moreover, a much more extensive characterization of the organoids is requiring, including histopathological assessment, marker expression and localization. In particular of the markers

that the authors pick to determine their subpopulations. Finally, one is least wondering why the authors go through the trouble to generate organoids if they do not use them to their full potential for, to the least, in vitro validation of their hypotheses.

We appreciate the Reviewer’s comment regarding the tumor contents in the tissue and organoid samples. Unfortunately, due to the low volume of tissues samples that are left from the current study, we were only able to perform WES on three patients and compared the CNV events with the inferCNV estimation from scRNA-seq samples. For these three patients, we sequenced both the paired tumor and normal samples, and performed somatic CNV analysis. Separately, we used non-tumor cells as reference to estimate the CNV events of the tumor cells in our scRNA-seq datasets using inferCNV.

In summary, we added the following description in the revised manuscript “We also performed whole exome sequencing (WES) for three RP samples to confirm tumor content. In the WES samples, we detected somatic mutations in known PCa genes, including *KMT2D*, *MTOR*, *SPOP*, and *PIK3R1* (Supplemental Table 4), indicating tumor content in tissues assessed by single cell analysis. Copy number analysis of the WES samples revealed consistent CNV events with the inferCNV estimation, such as chr4q31 amplification, chr11q24 amplification, and chr19q13 deletion (Supplemental Table 4), supporting our tumor cell identification.”

For the organoid samples, unfortunately we did not have sufficient remaining samples for further DNA sequencing studies. When we ran the copy number estimation test on the organoid cells, we found that using the non-tumor cells as reference, tumor cells were estimated to harbor significantly different CNV profiles. Even though this copy number estimation was still based on the expression matrices of the samples, it provides support for our results that we profiled tumor-like cells in one particular organoid sample. We were surprised that we would find putative tumor cells in organoids from primary prostate cancer samples. We acknowledge that the identity of tumor cells cannot be definitively determined by gene expression profiles, on a transcriptomic level, we found that this “tumor-like” population was similar with the previously established *ERG+* tumor cell, showing significantly higher expression of *ERG*, *PCA3*, and *AMACR* (see below).

In summary, we acknowledge that the annotation of these putative tumor cells requires further experimental validation in the future, which is a limitation for our current study and beyond the scope of this study. Therefore, we have revised the discussion as follows “We detected a population of “tumor-like” cells largely from one patient organoid with significantly different CNV profiles compared to other organoid epithelial cells. It is possible that these “tumor-like” cells are present only initially in some organoids but

dissipate after passage”. Our major focus for the organoid section was that we observed basal cells and club cells in the organoids that showed the same markers as in the tissue samples, and that we detected cell-state differences in the organoid samples compared to their parent tissue samples.

The execution of the computational pipelines looks very standard, usage of Seurat, STAR Fusion and CNVInfer. My concern is, scRNA-seq suffers from shallow sequencing and the authors have used CNVInfer tool to look for CNV. The point is, small focal changes will not be captured as compared to WGS or exome. In addition, if one feeds in a list of expected copy number alterations from bulk RNA seq or WGS a better resolution/capture of CNV can be achieved (I did not see this part being mentioned).

We appreciate the Reviewer’s comments that using an expected CNV from bulk RNA-seq or WGS would be good validation of the tumor cells in our analysis. However, we and others have observed that inferCNV profiling does not necessarily represent the actual sites of copy number alterations but is also influenced by which cell types are used for reference. For instance, in the Karthaus et al. study, the authors performed an unsupervised inferCNV run for each PCa patient in the analysis without a reference group and found that most putative tumor cells harbored an amplification in chr8 and chr1. In the recent study on PCa scRNA-seq by Chen et al., using the non-malignant cells from the Henry et al. study as reference, it was shown that the tumor cells harbored copy number amplifications in multiple chromosomes including chr 8,9 and 11. Therefore despite the shallow sequencing of scRNA-seq, the estimated CNV pattern for the inferred tumor cells may be vastly different among datasets and different selections of reference.

To confirm tumor content, we have now performed WES on three paired tumor and normal samples from our dataset and showed that even though WES CNV results were not well aligned with the inferCNV estimation, CNV events such as chr4q31 amplification, chr11q24 amplification, and chr19q13 deletion, detected from the WES samples were also revealed by the inferCNV results, which provided evidence for our tumor cell identification.

In the revised manuscript, we added the following text in the Discussion section of the revised manuscript to acknowledge that some CNV might escape the detection of our analysis: “The identification of tumor cells in this study was largely based on the CNV estimation using the non-malignant epithelial cells as reference. Given the relatively shallow sequencing depth of scRNA-seq, we acknowledge that it was possible that small focal changes of CNV might not be well captured in our analysis”.

Reviewer #5 (Remarks to the Author): Expert in immunogenomics and single-cell RNA-seq

This work contains satisfactory cohort and scRNAseq data. The data generated for this analysis is sufficient to clearly determine that the ERG status has significant effect on the immune and stromal compartments. However, few comments needs to be addressed prior to publication:

Moderate:

The authors suggest that the stratification of the T-cell populations based on ERG status showed two CD4 T-cell clusters that were differentially enriched. While this is correct there are additional differences which were overlooked for example CD8 cluster 1 and CD8 cluster 2 shows different internal distribution. Thus, to avoid biases due to clustering granularity it would be recommended to perform additional re-clustering of these populations. Such analysis could directly affect the authors conclusion regarding the immune microenvironment in ERG+/-

We thank the Reviewer for the suggestion of clustering CD8 T-cells to investigate the composition difference. We have now subset the CD8 T-cells and performed a sub-clustering analysis and reached the same clustering as the overall T-cell analysis (see below). Within these three clusters, the largest cluster (CD8) represents a more cytotoxic and exhausted CD8 T-cell population due to the high expression of TIGIT and GZMA.

However, when computing the distribution of ERG+ and ERG- patients in these three clusters, we found that despite the difference of enrichments in these three clusters, they were not statistically significant by Fisher's Exact test. Therefore, our conclusion in the manuscript is still supported in that differential enrichments based on ERG status.

In addition, the authors claim that CD4 and CD8 T-cells associated with ERG- tumor cells represented a more exhausted and cytotoxic phenotype, yet they show only a couple of markers for these T-cell states. Are these exhausted or activated cells? An additional signature enrichment analysis will further strengthen their claim. Here also please note that - CCR7+, CD69: higher in ERG+, granzyme A and B: higher in ERG- and MKI67 is higher in ERG-.

To further illustrate the nature of CD4 and CD8 T-cells associated with ERG- tumor cells, here we have now tested T-cell activation markers including CD69, CD71 (TRFC), CD25 (IL2RA) and HLA-DR. In CD4 T-cells, CD69 expression was higher in ERG+ patients but the other three markers showed no significant differences. And none of these four

markers were differentially expressed in *CD8* T-cells (below). These comparisons are now included in the Supplemental Figure 7 d-e and we have added the following statements in the revised manuscript “However, when testing T-cell activation markers such as *CD69*, *TRFC*, *IL2RA*, and *HLA-DRA* in both *CD4* T-cells (Supplemental Figure 7d) and *CD8* T-cells (Supplemental Figure 7e), no statistical significance difference in expression was detected between *ERG+* and *ERG-* tumors, suggesting that *ERG* status was unlikely to be associated with higher T-cell activation”.

We agree and thank the Reviewer for suggesting testing another exhausted T-cell signature gene set to support our analysis (See below, Woroniecka et al., *Clinical Cancer Research*, 2018) on *CD4* and *CD8* T-cells. We detected a significant enrichment of this signature in *CD4* T-cells associated with *ERG-* patients, which was consistent with our findings in the manuscript that *CD4* T-cells in *ERG-* patients appear to be associated with more exhausted T-cells. Given space limitations, we are happy to include this additional analysis if the Reviewer prefers.

We thank the Reviewer for pointing out the difference in expression of markers such as *CCR7*, *CD69*, *GZMA*, *GZMB* and *MKI67*. Among these markers, *GZMA* and *GZMB* were both considered to be cytotoxic T-cell markers therefore consistent with our analysis that *ERG-* *CD4* T-cells were likely to be in a more cytotoxic and exhausted niche.

However, *MKI67* was only expressed in less than 20 cells in *CD4* or *CD8* T-cells therefore the difference was not statistically significant or representative to make the conclusion that *ERG-* *CD4* or *CD8* T-cells were in a more proliferative state. Therefore we summarized the comparison in the manuscript as “we tested the expression for these markers in both *ERG+* and *ERG-* *CD4* T-cells and found a significantly higher proportion of *CCR7+* central memory *CD4* T-cells, *GZMB+* cytotoxic *CD4* T-cells and *TOX+* exhausted *CD4* T-cells associated with *ERG-* patients”.

Finally, the analysis in 7e of the common pathways upregulated in cancer, *CD4* cells and stromal cells is not clear. In the T-cells, not only *CD4* cells show a difference, in the stromal cells they emphasized the fibroblast and the cancer cells have multiple clusters.

We agree with the Reviewer that not only *CD4* T-cells showed a composition difference. We also detected a difference of contribution in the fibroblast population. However, we performed the subclustering analysis in the tumor cells to illustrate the transcriptomic profile difference between *ERG+* and *ERG-* tumor cells. and in Figure 7, we were testing if the differences in the tumor cells between *ERG-* and *ERG+* patients could potentially drive distinct and common stromal and immune responses. Therefore we combined the tumor cell subclusters and stratified them only based on the *ERG* status. In the revised manuscript, we removed the term “cluster” when we described the stromal population to avoid any confusion as we did not observe multiple clusters in the fibroblast population.

Minor:

1. Fig 7a – Please provide the marker gene list used for the annotation (for specific T-cells subsets: *CD8+*, *CD4+* Treg).

In the T-cell analysis, we identified the different subsets using graphical clustering algorithm and annotated them using the top differentially expressed markers in each subset as well as known markers such as *CD4*, *CD8A*, *CD8B*, *GZMB* (NK-cell) and *FOXP3* (Treg). We have now included the top 50 differentially expressed markers for each subset in the Supplementary table 7.

2. Fig 7a – Please keep the same colors scheme *ERG-* (blue) and *ERG+* (red) in a,b,c,d. (it is currently different between the UMAP and bar-graphs).

We appreciate the thoroughness from the Reviewer and have now ensured the *ERG-* and *ERG+* colors are consistent in Figure 7a.

3. Line 419, I could not find the data for upregulating genes in *CD4+* cells (*FOS*, *JUN*, *CXCR6* etc.)

In the revised manuscript, we have added a reference to the supplementary table 7 where we listed the differentially expressed genes between the two *CD4* T-cell clusters (Results, “*FOSB* ($\log_2FC = 1.79$, $FDR q = 5e-30$), *FOS* ($\log_2FC = 1.78$, $FDR q = 6.2e-26$) and *JUN* ($\log_2FC = 1.55$, $FDR q = 5.5e-22$) (Supp Table 7).

4. Line 434 need to add fig 7d.

Fixed.

5. Line 437, It is not clear why the authors analyzed all stromal cells for the analysis, rather than just the fibroblast, which they have shown to have different distribution between ERG+ and ERG- (similar to CD4+ cells).

For the stromal cell analysis, even though we detected a different distribution in fibroblast population regarding *ERG* status, both the differential expression analysis and clustering analysis indicated that *ERG*+ fibroblast and *ERG*- fibroblast did not cluster separately such that their fibroblast gene expression profile was more dominantly homogeneous than the differences attributed to the *ERG* status, which could be due to a low cell detection issue. Therefore, to fully examine the effect of *ERG* status in the stromal population, we combined fibroblast, endothelial cell and smooth muscle.

6. In fig 7a as in fig 7c, please position the legends at the same place (for example in 7c it could confuse the readers to think that ERG- is in smooth muscles and ERG+ in fibroblast)

We thank the Reviewer for pointing this out. It is now fixed in the revised figure (Figure 7a and 7c).

7. Line 441: to change from fig 7f to e.
Fixed.

8. Line 444: to change from 7g to f.
Fixed.

9. Line 448: to change from 7h to g.
Fixed.

0. Supplemental fig 6 – Please change the mode of visualization, or add mean, so we can visualize up or downregulated genes, in the current violin plot version, we cannot appreciate changes in genes that are not described in the text. It would be interesting to know if the markers of the same family without significant differences are increased or decreased in the same way (ex: *HAVCR2*, *LAG3*, *Tigit* in CD4+)

We apologize for the unclear visualization. In the revised figure (Now Supplemental Figure 7) we have edited the figures such that for each violin plot, a blue line was added to illustrate the mean expression level of the marker.

We have now updated Supplemental Figure 7 to include markers in the same family such as *CTLA4*, *HAVCR2*, *LAG3*, and *TIGIT*, for *CD4* and *CD8* T-cells. In the *CD4* T-cell comparison, the mean expression levels of all these four markers were found to be higher in *ERG*- *CD4* T-cells but only *CTLA4* showed a statistically significant difference ($p < 0.001$). In the case of *CD8* T-cells, the mean expression levels of all these four markers were also found to be higher in *ERG*- *CD8* T-cells but only *HAVCR2* and *LAG3* met statistical significance. We thank the Reviewer for the suggestion of adding the mean expression such that even without significant differences, this comparison further supported our interpretation that *ERG*- status might give rise to a more exhausted T-cell niche.

1. Supplementary fig 7d and e: be consistent with the colors of the legends (M0, M1, M2 macrophages)

We thank the Reviewer for pointing this out. We have fixed this in the revised figure (Supplemental Figure 8d) and the color selections are consistent for the three types of macrophages.

REVIEWERS' COMMENTS

Reviewer #1 (Remarks to the Author):

The authors have correctly responded to the questions and suggestions.

Reviewer #2 (Remarks to the Author):

The authors were graciously responsive to my comments. I think this will be an excellent addition to the field.

Douglas Strand, PhD

Reviewer #3 (Remarks to the Author):

My comments on the organoid section have been answered in a satisfactory fashion

Reviewer #4 (Remarks to the Author):

The authors have addressed my concerns.

Reviewer #5 (Remarks to the Author):

The authors have addressed all of my concerns and comments. The revised manuscript is ready for publication.